# GAIN: ON THE GENERALIZATION OF INSTRUCTIONAL ACTION UNDERSTANDING

**Junlong Li**[1], **Guangyi Chen**[2,3], **Yansong Tang**[1], **Jinan Bao**[4],
**Kun Zhang**[2,3], **Jie Zhou**[1], **Jiwen Lu**[1,*]
[1]Tsinghua University, [2]MBZUAI, [3]Carnegie Mellon University, [4] University of Alberta

## ABSTRACT

Despite the great success achieved in instructional action understanding by deep learning and mountainous data, deploying trained models to the unseen environment still remains a great challenge, since it requires strong generalizability of models from in-distribution training data to out-of-distribution (OOD) data. In this paper, we introduce a benchmark, named **GAIN**, to analyze the Generaliz-Ability of INstructional action understanding models. In GAIN, we reassemble steps of existing instructional video training datasets to construct the OOD tasks and then collect the corresponding videos. We evaluate the generalizability of models trained on in-distribution datasets with the performance on OOD videos and observe a significant performance drop. We further propose a simple yet effective approach, which cuts off the excessive contextual dependency of action steps by performing causal inference, to provide a potential direction for enhancing the OOD generalizability. In the experiments, we show that this simple approach can improve several baselines on both instructional action segmentation and detection tasks. We expect the introduction of the GAIN dataset will promote future in-depth research on the generalization of instructional video understanding. The project page is `https://jun-long-li.github.io/GAIN`.

## 1 INTRODUCTION

Instructional videos play an essential role for learners to acquire different tasks. The explosion of instructional video data on the Internet paves the way for learners to acquire knowledge and for computer vision community training models, for example, human can train an action segmentation model to understand the video by the dense step prediction of each frame, or an action detection model to localize each step. While a number of datasets for instructional action understanding (IAU) have been proposed over the past years(Alayrac et al., 2016; Das et al., 2013b; Malmaud et al., 2015; Sener et al., 2015) and growing efforts have been devoted to learning IAU models(Zhukov et al., 2019; Huang et al., 2017), the limited generalizability of models remains to be a major obstacle to the deployment in real-world environments. One may ask a question "*Suppose the model has learned how to inflate bicycle tires, does it know how to inflate car tires?*" In fact, due to potential environmental bias between the training dataset and application scenes, the well-trained model might not be well deployed in an OOD environment (Ren et al., 2019), especially when instructional videos of interest to users are not involved in the finite training dataset.

To encourage models to learn transferable knowledge, it is desirable to benchmark their generalizability. Though this OOD generalization problem (Barbu et al., 2019; Hendrycks et al., 2021; Hendrycks & Dietterich, 2019) attracts much attention in the field of image recognition, such as ObjectNet (Barbu et al., 2019) and ImageNet-R (Hendrycks et al., 2020), it has barely been explored for the IAU task. A related problem is video domain generalization (Yao et al., 2021) (VDG) for conventional action recognition which focuses on domain generalization when changing the scene or background of the action. However, different from conventional action, the key obstacle to the generalization of instructional action is the distribution shift of action steps under different task categories, which is caused by the collection bias of the datasets. In Fig. 3, we show that the steps under different task categories have different distributions.

---

[*]Corresponding author

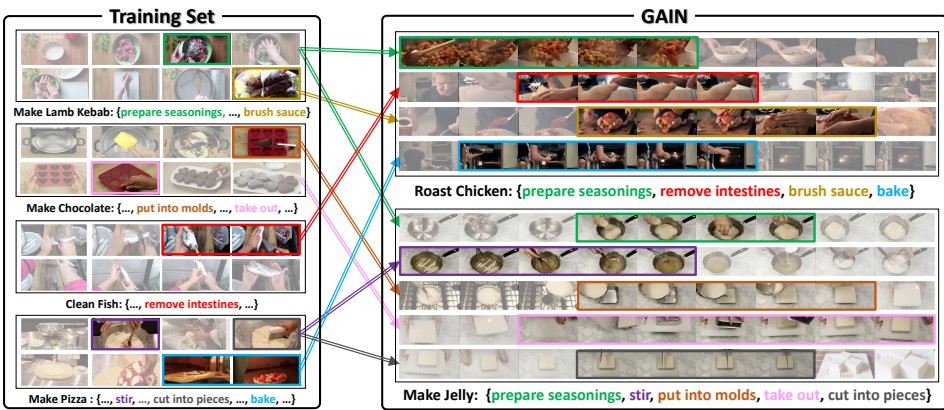

Figure 1: Two examples of constructing new OOD instructional tasks by reassembling the steps of in-distribution videos in training datasets. For example, the OOD task "Make Jelly" consists of five steps: $\{prepare\ seasonings, stir, put\ into\ molds, take\ out, cut\ into\ pieces\}$, where the "prepare seasonings" step is in the task "Make Lamb Kebab", the "put into molds" and "take out" steps come from "Make Chocolate", and the "stir" and "cut into pieces" steps are in the task "Make Pizza". The steps in GAIN are consistent with those in the training set, with non-overlapping task categories. GAIN encourages models to transfer the knowledge learned from training data for OOD data.

Given the motivation that action steps are the key research objects of IAU and have distribution shift when task categories change, we propose a new evaluation strategy to benchmark the generalizability by re-constructing test task categories using the steps of training tasks and evaluating the models with these new task categories. In the reconstruction, we require that training and testing task categories are different but step categories are consistent. As shown at the bottom of Fig. 1, we try to find a new testing task "Make Jelly" with existing step categories including the "prepare seasonings" step in the task "Make Lamb Kebab", the " put into molds" and "take out" steps in "Make Chocolate", and the " stir" and " cut into pieces" steps in "Make Pizza". This construction is non-trivial since existing IAU datasets cannot be directly used. First, for most IAU datasets (such as COIN (Tang et al., 2019)), the steps in different videos are not shared, therefore, we cannot construct testing data by splitting itself. Second, though CrossTask (Zhukov et al., 2019) also collects cross-task videos with partial steps shared, these shared parts are only a minority in the dataset (only 14% steps are shared, i.e. 73 are shared of a total of 517 steps) and most videos have steps that are not shared with others. Besides, because the related tasks are not fine-grained annotated, they cannot be used for evaluation. Furthermore, it is built to investigate whether sharing constituent components improves the performance of weakly supervised learning. It motivates us to collect and annotate a real-world IAU dataset, **GAIN**. It consists of 1,231 videos of 116 OOD tasks with 230 categories of steps, covering a wide range of daily activities. All videos in our GAIN dataset are employed for evaluation. These videos can be split into two groups: GAIN-C and GAIN-B, as counterparts of the COIN (Tang et al., 2019) and Breakfast (Kuehne et al., 2014) datasets, respectively.

Furthermore, we propose a simple yet effective approach to enhance the generalizability of IAU models by cutting off excessive contextual dependency by performing causal inference. It is inspired by the observation that model generalizability is inevitably influenced by short-cutting with a biased context. Compared with previous methods learn the temporal dependency among video steps by the temporal networks, such as TCN (Lea et al., 2017) applying a hierarchy of temporal convolutions, we propose to reduce the over-dependency between steps to mitigate the negative effect from temporal context bias. For example, if the task "Inflate Bicycle Tires" is always observed together with the "bicycle pumps" during the training process, this knowledge will be difficult to transfer to the OOD task "Inflate Car Tires" with other inflaters. In our approach, we apply the Back-Door Criterion to infer causal effect, and present a Monte Carlo based method to approximate the distribution after "intervention". The method is evaluated with various baseline methods on both action segmentation and detection tasks, and is shown to produce consistent improvements.

**Contributions.**   (1) We propose a new evaluation strategy to benchmark the generalizability of IAU models by evaluating the models on the OOD tasks. (2) We build a real-world OOD instructional video dataset, **GAIN**, where the OOD tasks are constructed by reassembling the steps of

training datasets. (3) We propose a simple yet effective approach, cutting off excessive contextual dependency by causal inference, which provides a potential direction to enhance generalizability.

## 2 THE GAIN DATASET

In this section, we introduce our GAIN dataset, a video-based dataset covering a large range of daily tasks reassembled via a specific framework, which collects the tasks whose categories are different from training tasks to benchmark the generalizability of IAU models. For convenience, we call the tasks whose categories are same with the tasks in the training dataset as in-distribution tasks, and the ones with different categories as OOD tasks. Note that, when we mention "in-distribution" and "OOD", the variables are steps but not tasks. It means the steps are "in-distribution"/ "OOD" under same/different tasks. To our best knowledge, GAIN is the first dataset to evaluate the generalizability of IAU models on the OOD steps. Fig. 2 shows the pipeline to construct the GAIN dataset. Below, we describe the details of our dataset, including how to benchmark the generalizability, how to collect the data and construct the dataset, and the basic dataset statistics.

### 2.1 PROBLEM DEFINITION

The generalizability is of critical importance to the models for the deployment in a real-world environment, especially for IAU systems, e.g., we expect the model can know how to "Inflate Car Tires" after learning how to "Inflate Bicycle Tires". For this goal, we propose to benchmark the generalizability of IAU models by building an OOD evaluation dataset, in which task categories is constructed by reassembling the steps of the training set. With this construction setting, the step categories are consistent and step distribution is changed.

The training dataset $\boldsymbol{X}^T = \{X_i^T\}_{i=1}^{n_T}$ contains $n_T$ instructional videos $X^T$, where each video is composed as a set of steps $X_i^T = \boldsymbol{S}_i^T$. This step set can be formulated as $\boldsymbol{S}_i^T = \{s_{i,j}^T\}_{j=1}^{n_s}$, where $n_s$ is the number of steps in a video. In the conventional experimental setting, both training and evaluation data are in-distribution, which is formulated as $\boldsymbol{X}^T \overset{i.i.d}{\sim} \mathcal{X}_{Source}$ and $\boldsymbol{X}^E \overset{i.i.d}{\sim} \mathcal{X}_{Source}$, where $\mathcal{X}_{Source}$ denotes the data distribution and $\boldsymbol{X}^E$ denotes the videos in the evaluation set. To benchmark the generalizability of models, we collect the videos of unseen tasks with seen steps, where videos in the evaluation set are OOD tasks that can be formulated as follows:

$$\boldsymbol{X}_{OOD}^E \overset{i.i.d}{\sim} \mathcal{X}_{Target}, \qquad \boldsymbol{X}^T \overset{i.i.d}{\sim} \mathcal{X}_{Source} \tag{1}$$
$$s.t. \quad \Omega_S^T = \Omega_S^E,$$

where $\Omega_S^T$ and $\Omega_S^E$ respectively denote the set of all steps in the training and evaluation set, and $\boldsymbol{X}_{OOD}^E$ denotes the collected OOD evaluation dataset. As shown in Fig. 1, we show some collected videos of our GAIN dataset, where the collected videos follow different step distributions but share the same step space. Finally, we evaluate the models trained on $\boldsymbol{X}^T$ with the OOD evaluation dataset $\boldsymbol{X}_{OOD}^E$ to benchmark the OOD generalizability.

Here we distinguish our defined OOD generalizability evaluation from other evaluation strategies. We summarize the comparisons with different evaluation methods in Table 1. First, compared to conventional supervised and unsupervised methods, we focus on the OOD evaluation to benchmark the generalizability of models. Second, UDA (Busto et al., 2018; Zhang et al., 2019) aims to transfer the knowledge from source domain to some known target domain. It needs the domain index (e.g. the target data) for the training process to minimize the domain gap between the source

Table 1: Comparisons of different evaluation strategies. **IV**: instructional video; **SL**: using source label during training; **TD**: using target data during training; **SC**: steps are consistent.

| Methods | IV | SL | TD | OOD | SC |
|---|---|---|---|---|---|
| Supervised (Tang et al., 2019) | ✓ | ✓ | ✗ | ✗ | ✓ |
| Unsupervised (Miech et al., 2019) | ✓ | ✗ | ✗ | ✗ | ✓ |
| UDA (Busto et al., 2018) | ✓ | ✓ | ✓ | ✓ | ✗ |
| Zero Shot (Sener & Yao, 2019) | ✓ | ✓ | ✓ | ✓ | ✗ |
| VDG (Yao et al., 2021) | ✗ | ✓ | ✗ | ✓ | ✗ |
| **Ours** | ✓ | ✓ | ✗ | ✓ | ✓ |

and known target. Compared with UDA, our OOD generalization further considers how to solve the problem without any domain indices. Though zero-shot recognition (ZSR) (Sener & Yao, 2019; Wang et al., 2019) also focuses on the generalizability of models, it is too difficult to conduct zero-shot analysis directly for IAU models since ZSR requires the models to understand unseen action steps. This setting can be used for task-level actions (e.g. classification) given extra descrip-

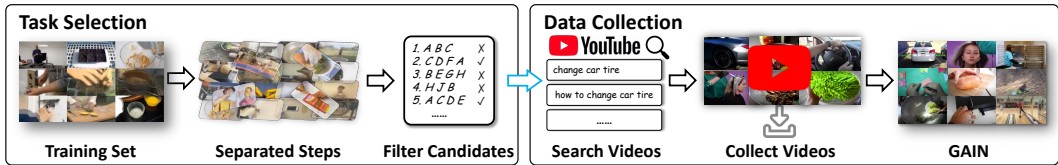

Figure 2: The pipeline to construct GAIN, which includes Task Selection (left) and Data Collection (right). Given an instructional video training set, we first separate the steps of these tasks and generate a large number of task candidates. Secondly, we filter out the unqualified ones according to three principles. Then, we search for YouTube videos related to the selected tasks and download the videos, which embrace high relevance with queries, explicit instructions, and rich diversity.

tions (Wang et al., 2019), but not for complex action understanding tasks such as action segmentation or action detection. Recently, many methods in the field of image classification have attempted to evaluate the generalizability of models by collecting or generating the OOD data, e.g. Object-Net (Barbu et al., 2019) and ImageNet-R (Hendrycks et al., 2021). However, how to evaluate the generalizability of models for more complex IAU task has barely been visited. The most related one is VDG (Yao et al., 2021), which evaluates the domain generalization ability of action recognition models when changing the scene or background. Unlike VDG, our setting focuses on the distribution shift of action steps when task categories are changed in the target domain, which is more common in the field of IAU.

## 2.2 Task Selection

To construct an evaluation dataset consisting of diverse and high-quality daily tasks, we choose the largest fine-grained annotated dataset, COIN (Tang et al., 2019), and the widely-used instructional video dataset, Breakfast (Kuehne et al., 2014), as the training sets.

How should we select the new tasks to benchmark the generalizability? We argue that the tasks in our GAIN dataset require three basic principles as follows:

- **Task Non-overlapping:** The steps in our GAIN dataset are out-of-distribution, which requires the tasks in GAIN to be non-overlapping with those in the original training set. The model performance on these non-overlapping tasks can intuitively indicate the generalizability.
- **Step Consistent:** Despite the step distributions are different under non-overlapping tasks, we require that the categories of steps in the testing videos are consistent with the training dataset. On the one hand, with totally different steps, IAU will be even more difficult, which deflects our goal to benchmark the generalizability. On the other hand, the steps in the training set are common in daily life, which is of critical importance for IAU.
- **Category Diverse:** The third principle, category diverse, encourages the annotators to discover more diverse data. In other word, we argue that the larger number of task categories is the better. For example, a dataset (with 3 tasks) contains 2 videos of repairing a car, 2 videos of repairing a roof and a video of repairing a television is more diverse than a dataset (with only 1 task) with 5 videos of repairing a car. More diverse data indicates more reliable benchmarking.

With the principles above, as shown in Fig. 2, we first generate a large number of task candidates (i.e. step combinations) and then filter out the unqualified ones. Specifically, we apply the steps in the training dataset as the anchor steps and generate 10 step combinations with the caption clues, where each step combination contains 2∼5 steps. The captions in instructional videos often mention steps that are not in the current task but closely related to other steps in the video, which could help to generate step combinations. In total, we generate more than 8,000 task candidates.

Here we provide details to show how we filter the candidates. Given a candidate, we first check whether it is logical. For example, if the candidate is "lifting jack, replace the tire, remove the jack", it makes sense because it could form a task "Change Car Tire"; it is not acceptable if the candidate is "lifting jack, replace the tire, add the seasoning", which may never happen sensibly in daily life. Then according to Task Non-overlapping, we drop the logical candidates if they make up tasks that already exist in training sets. Since all candidates are composed of steps directly from training sets, they inherently satisfy the principle Step Consistent.

In the first round of annotation, we ask 11 annotators to go through these candidates and annotate whether the candidate is reasonable and satisfies the above principles. We filter out the candidates

annotated as unqualified by more than half annotators and finally select 147 candidates. In the second round, annotators are asked to name the new OOD task, refine current step combinations, and collect the videos from the Internet. By filtering out the rare actions, we finally collect 1,231 videos of 116 tasks. In the last round, the annotators label the fine-grained temporal boundaries of each steps in videos.

## 2.3 DATA COLLECTION

Given the selected tasks, we search for YouTube videos related to the task names. We use a query with exactly the task name or the task name following a "how to" prefix to locate instructional videos, e.g. for the task "Change Car Tire" we use "change car tire" or "how to change car tire". To improve the quality and diversity, we adopt several criteria to select videos including: high relevance with queries, explicit instructions, and rich diversity. We prefer videos more relevant to the queries and containing explicit instructions with pictures, since visual models are not able to only learn from narrations. Besides, videos with explicit steps to complete tasks are also favorable, although there might be no vocal instructions. Furthermore, explicit steps in a video do not need to exactly match those in its task – in other words, permuting and being a proper subset of the task are acceptable. Moreover, if similar steps are witnessed in differ-

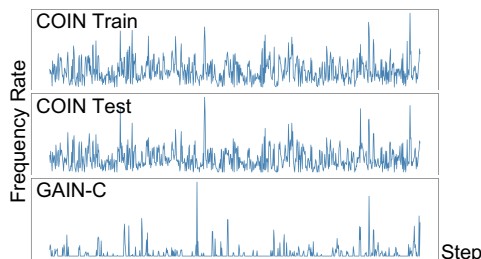

Figure 3: The step distributions of the training dataset, original in-distribution test dataset, and our OOD test dataset on COIN. Under the same task categories, the step distribution is similar to the original training and test datasets. With different task categories in GAIN-C, the step distribution changes a lot, which supports our assumption that steps are in-distribution/OOD with same/different task categories.

ent videos of a task, like "add salt" and "add sugar", we regard them as the same step. With regard to the undefined steps, they are considered as the background and not further annotated. On the one hand, during the data collection stage, if a video contains a long stretch of undefined but important steps, this video will be filtered out according to the principle of Step Consistent. On the other hand, videos with undefined meaningless steps are acceptable and these steps will be considered as the background. After collecting the videos, we utilize the annotation tool provided in (Tang et al., 2019) to label the corresponding step categories and segments.

## 2.4 STATISTICS

The final version of the GAIN dataset consists of 1,231 instructional videos related to 116 unseen tasks. Our GAIN dataset is a pure evaluation dataset to benchmark the generalizability of IAU models. Each task in GAIN contains 2∼24 videos with an average of 10 videos. We annotate 6,382 action segments in GAIN with an average of 5 steps in each video. The average length of videos is 2 minutes and 30 seconds, and the average length of steps is 12 seconds. Totally, the GAIN dataset contains OOD videos of 51.2 hours for generalizability evaluation.

GAIN can be divided into two splits as counterparts of COIN (Tang et al., 2019) and Breakfast (Kuehne et al., 2014) datasets, and we name them **GAIN-C** and **GAIN-B**, respectively. COIN is a large-scale benchmark with 9,030 training videos and 2,797 testing videos of 180 tasks. As its counterpart, GAIN-C contains 1,000 videos of 100 unseen tasks with a length of 41.6 hours in total, where 5,238 segments are annotated. Fig. 3 shows the step distributions on COIN Train, COIN Test, and our GAIN-C, where horizontal axis denotes the different steps and the vertical axis denotes the frequency rates of these steps. We can observe the step distributions in original COIN Train and Test sets are similar, but different from our GAIN-C. It demonstrates our assuming that under different task categories, the step distributions are different. Breakfast is composed of more than 1.9k cooking-related videos of 10 breakfast routines such as "Make Coffee" and "Cook Pancakes". Accordingly, the GAIN-B split includes 231 videos of 16 OOD tasks with an average length of 2 minutes and 30 seconds. These tasks consist of 20 fine-grained action categories. We provide more statistical data and analysis in the **Appendix**.

## 3 METHOD

In this section, we first construct a causal graph for the IAU problem. Then, we introduce our method which applies causal inference to mitigate the negative effect of confounding context bias.

### 3.1 Causal Graph Construction

The widely researched action understanding tasks include action segmentation and detection, which both focus on the steps. Without loss of generality, we formulate the task as:

$$P(Y|S) = f_\theta(S), \tag{2}$$

where $S$ denotes a step in the video $X$, $Y$ is the prediction and $f_\theta$ represents the model. Then, we formulate the action understanding framework in light of a causal graph $\mathcal{G} = \{\mathcal{V}, \mathcal{E}\}$, where the nodes $\mathcal{V}$ include the step $S$, model prediction $Y$, and context steps $Z$. Note that $X = S \cup Z$ and $S \cap Z = \emptyset$ denote video $X$ can be divided by query step $S$ and context steps $Z$. The links $\mathcal{E}$ indicate the dependence (computational but not strict causal direction (Liu et al., 2021)) between two variables. For example, $S \rightarrow Y$ in the causal graph indicates variable $S$ is the cause of variable $Y$. We show the casual structure of the IAU problem in Fig. 4(a) and explain it as follows:

- $Z \rightarrow Y \leftarrow S$ indicates that the model prediction depends on both the step $S$ and the context steps $Z$. For example, when recognizing the current step $S$, temporal models (e.g. LSTM (Hochreiter & Schmidhuber, 1997) and C3D (Tran et al., 2015)) always use the temporal context clues for current prediction, which leads to $Z \rightarrow Y$.
- $S \leftarrow Z \rightarrow Y$ denotes that the video context steps $Z$ simultaneously affects the steps and model prediction. $Z \rightarrow Y$ has been explained above and $S \leftarrow Z$ is intuitive due to the temporal dependency of video. Thus we call the context bias $Z$ is a confounder (Pearl, 2009), which misleads the model to focus on the spurious correlation, reducing the generalizability of the model. The casual graph describes the information flow during the inference. When $S$ is being estimated, other context steps are $Z$, and since $S$ is affected by $Z$ during the inference, $Z$ points to $S$.

Then we show that the model prediction is misled by the spurious correlations of context bias when we only consider the likelihood $P(Y|S)$. As shown in Fig. 4(a), we re-write $P(Y|S)$ with the Bayes rule as:

$$P(Y|S) = \Sigma_{\boldsymbol{z}} P(Y|S, Z = \boldsymbol{z}) P(Z = \boldsymbol{z}|S), \tag{3}$$

which denotes that the likelihood $P(Y|S)$ are influenced by $P(Z = \boldsymbol{z}|S)$. Now we use an example to show that $P(Z = \boldsymbol{z}|S)$ is biased. In the video "Inflate bicycle tires" , current content $S$ is "installing the nozzle" and the context $Z$ is "using bicycle pump". The content $S$ and the context $Z$ are always observed together in the training process and thus $P(Z = using\ bicycle\ pump|S = installing\ the\ nozzle)$ is higher. It leads the model to predict higher probability $P(Y|S = installing\ the\ nozzle)$ when observing the $Z = using\ bicycle\ pump$ and vice verse. However, when we apply the model to analyze the OOD video "Inflate car tires", where $Z$ "using bicycle pump" is absent, the model may be confused and consequently give wrong prediction. Additionally, although there is actually bidirectional effect between $S$ and $Z$, we find that Bayes' theorem and Eq.3 remains unchanged. In Fig. 4(a), we apply $S \leftarrow Z$ to highlight the bias caused by the con-fonder $S \leftarrow Z \rightarrow Y$, which is demonstrated by the Fig. 3.

Motivated by the causal inference method (Pearl, 2009; Glymour et al., 2016), we propose to conduct intervention to alleviate the negative effect of context bias. In causal inference, the intervention is represented as $do(\cdot)$. Once intervened, a variable will have no in-coming links anymore and the previous in-coming links in the causal graph are cut off. As shown in Fig. 4(a), when we intervene $S$ with the Back-Door Criterion in (Pearl, 2009), i.e. $do(S)$, the link between $S$ and $Z$ is cut off so as the dependency. We formulate the model prediction process under the intervention:

$$P(Y|do(S)) = \Sigma_{\boldsymbol{z}} P(Y|S, Z = \boldsymbol{z}) P(Z = \boldsymbol{z}), \tag{4}$$

where $Z = z$ is independent from $S$. Thus, after intervention, when the model predicts from $do(S)$ to the label $Y$, it fairly takes every $z$ into consideration. Please see more detailed introduction and derive about Back-Door Criterion in Section 3.3 of (Glymour et al., 2016).

However, the intervention is a great challenge, since the prediction under this intervention is subject to the prior $P(Z)$, which is difficult to compute numerically. Thus, we simulate conducting the intervention. We replace the numeric process with a sampling process and approximate the prior $P(z)$ by the Monte Carlo method. As shown in Fig. 4(b), we regard each step as an individual instance and put all the steps in a lottery box, i.e. a step pool. Statistically, all the steps in the training set form the population $Z = \{z_1, ..., z_n\}$ with $n$ categories, and then we can sample from this population. During the sampling process, the frequency of $z$ is not affected by $X$ anymore and is

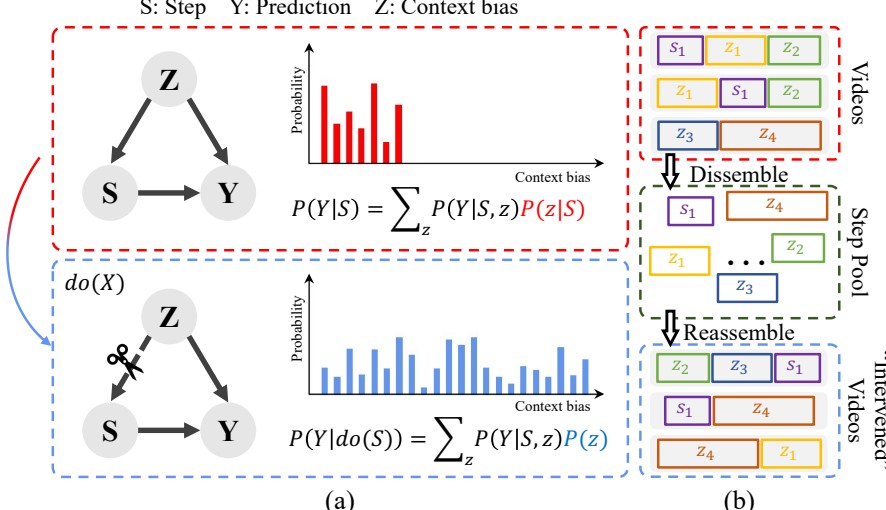

Figure 4: **(a)** The causal inference illustration for instructional action understanding. (Top) presents the original causal graph of IAU and the likelihood $P(Y|S)$. (Bottom) shows the causal graph and the causation $P(Y|do(S))$ after intervention. **(b)** Approximation with Monte Carlo method. We first dissemble the videos, where $s_1$ only occurs with $z_1 \& z_2$, and sample from the step pool. The prior $P(Z)$ is approximated with the relative frequency and sampled steps constitute the "intervened" videos, where $s_1$ could be observed with $z_3$ or $z_4$.

only related to the statistics of the training set. We approximate the prior with the relative frequency:

$$P(Z = z') = \frac{\Sigma_{z \in \Omega} \mathbb{I}(z = z')}{\|\Omega\|}, \tag{5}$$

where $\Omega$ denotes the sampling population, $\|\Omega\|$ is the sample size, and $\mathbb{I}$ is an indicator function. We use the sampled steps to assemble "intervened" videos to learn causations of $X$ on $Y$, instead of the spurious correlations due to context bias $Z$. Fig. 4(b) illustrates an example of this process, in which $s_1$ only occurs with $z_1 \& z_2$ in the original videos and consequently models tend to learn spurious correlations of them. After dissembling and reassembling, in the "intervened" videos, $s_1$ could be observed with others, like $z_3 \& z_4$, and the occurrence of $z$ is not dependent on $s_1$. Technically, our causal "intervention" can be regarded as a new kind of data argumentation, where we dissemble the steps and reassemble them as new video data.

## 4 EXPERIMENTS

In this section, we provide performance comparisons between the in-distribution dataset and out-of-distribution GAIN dataset, and assess the effectiveness of our causal approach on both action segmentation and action detection tasks. We conduct experiments on three datasets, where COIN (Tang et al., 2019) and Breakfast (Kuehne et al., 2014) are used for both training and testing, and our GAIN dataset is only used for evaluation. As mentioned in Section 2.4, COIN and Breakfast are widely used for IAU, so GAIN is split into two groups as counterparts of them, named GAIN-C and GAIN-B respectively. Please refer to Section 2.4 for more descriptions of these datasets. The implementation details, results, and analysis are described below. More experimental results and visualization examples can be found in **Appendix**.

### 4.1 IMPLEMENTATION DETAILS

We conduct experiments with the following baseline methods: (1) LSTM(Hochreiter & Schmidhuber, 1997) is one of the earliest and most popular deep models dealing with temporal modeling. (2) ED-TCN(Lea et al., 2017) applies a hierarchic encoder-decoder framework with temporal convolutions, pooling, and upsampling to learn temporal patterns. We use 5 convolution layers for both the encoder and decoder, whose convolutional filters' sizes are 25. (3) TResNet(He et al., 2016) adds a residual stream in the encoder-decoder framework. We follow the network structure depicted in (Lei & Todorovic, 2018) and adopt the same experimental setting as ED-TCN's. (4) MS-TCN++(Li et al., 2020) proposes a multi-stage architecture, which first generates initial predictions and refines them several times. Our implementation is built upon the publicly provided codebase.

Table 2: Evaluation on COIN/GAIN-C with baselines and finer results across domains. · / · denotes performances on COIN/GAIN-C. C means we apply causal-based method.

| Domain | LSTM | C-LSTM | ED-TCN | C-ED-TCN | TResNet | C-TResNet | MS-TCN++ | C-MS-TCN++ |
|---|---|---|---|---|---|---|---|---|
| Nursing | 64.3/59.8 | 65.6/61.6 | 61.0/60.7 | 64.1/61.1 | 62.6/59.5 | 64.5/60.0 | 65.6/62.1 | 68.2/62.1 |
| Vehicle | 61.9/60.7 | 62.6/62.8 | 59.2/63.7 | 59.9/67.2 | 60.6/64.2 | 59.3/66.1 | 62.9/63.8 | 63.0/66.9 |
| Leisure | 58.0/60.4 | 59.3/62.9 | 55.1/60.4 | 56.9/61.8 | 54.9/60.0 | 57.0/60.1 | 61.8/63.3 | 61.2/62.2 |
| Gadgets | 67.4/60.3 | 68.6/63.9 | 64.9/63.2 | 66.4/66.5 | 64.2/63.9 | 65.6/64.2 | 68.3/61.4 | 68.2/64.7 |
| Electrical | 64.5/44.6 | 65.9/48.8 | 63.7/47.5 | 62.7/47.8 | 63.9/44.6 | 63.2/49.0 | 66.2/44.4 | 65.4/45.3 |
| Furniture | 61.5/57.0 | 63.5/59.1 | 58.8/59.3 | 59.7/58.3 | 60.4/60.2 | 60.9/60.4 | 63.8/60.0 | 63.6/61.8 |
| Science | 61.4/41.5 | 61.9/45.8 | 56.5/37.7 | 57.8/42.9 | 58.0/42.1 | 58.4/41.7 | 61.6/38.8 | 63.1/47.7 |
| Pets | 61.0/52.0 | 64.9/54.2 | 61.9/48.7 | 63.5/52.7 | 61.5/51.2 | 61.6/48.6 | 63.7/52.2 | 64.6/52.0 |
| Drink | 65.1/50.6 | 66.8/53.5 | 61.9/50.3 | 62.5/54.4 | 61.8/48.8 | 62.1/53.5 | 67.1/46.1 | 66.5/54.2 |
| Sport | 69.8/63.5 | 73.4/63.8 | 68.1/63.6 | 69.0/67.1 | 66.8/65.3 | 66.9/50.3 | 72.0/49.5 | 70.9/51.6 |
| Dish | 70.3/47.3 | 71.8/51.3 | 64.5/48.8 | 66.9/50.9 | 65.7/47.5 | 67.8/66.1 | 71.8/59.4 | 70.3/66.7 |
| Housework | 62.2/61.4 | 62.7/59.6 | 60.6/61.6 | 59.9/61.0 | 59.4/59.0 | 60.8/61.1 | 61.5/59.8 | 64.2/61.9 |
| Overall | 63.9/52.3 | 65.4/55.1 | 61.2/52.7 | 62.2/55.3 | 61.6/52.4 | 62.2/54.6 | 65.5/51.8 | 65.6/56.0 |

For COIN/GAIN-C, we use the temporal video resolution at 10 fps, and extract S3D(Miech et al., 2020) features with a pretrained model on HowTo100M (Miech et al., 2019) as the model input. And for Breakfast/GAIN-B, we use I3D (Carreira & Zisserman, 2017) features (pretrained on Kinetics (Carreira & Zisserman, 2017)) sampled at 15 fps as the model input. As for the original evaluation set, we follow the default setting and present results on split 1. For all experiments, we employ a $1 \times 1$ convolution layer to project the features into an embedding space, whose dimension is 64. Then, we apply different baseline methods to model the spatio-temporal clues. All settings are the same with both the baseline methods and our methods.

## 4.2 ACTION SEGMENTATION

**Setting:** Action segmentation aims at assigning each video frame with a step label. This task is a key step to understand complex actions in instructional videos. We adopt frame-wise accuracy, which is the number of correctly predicted frames divided by the number of total video frames. For Breakfast/GAIN-B, we also adopt edit distance and F1 score (Lea et al., 2017) at overlapping thresholds 10% to further measure the quality of the model prediction.

**Results:** The frame accuracy on COIN/GAIN-C is shown in Table 2 (the Overall row). We observe an obvious performance drop of approxi-

Table 3: Results on Breakfast/GAIN-B for action segmentation.

| Method | ED-TCN | C-ED-TCN | TResNet | C-TResNet | MS-TCN++ | C-MS-TCN++ | LSTM | C-LSTM |
|---|---|---|---|---|---|---|---|---|
| Acc. | 42.4/21.3 | 44.5/35.4 | 49.1/18.0 | 49.2/30.8 | 67.6/16.8 | 66.4/32.1 | 47.8/25.5 | 45.7/37.9 |
| F1@10 | 37.8/3.2 | 42.7/5.3 | 41.6/3.7 | 43.0/5.5 | 57.1/3.1 | 54.7/3.7 | 4.0/0.3 | 4.8/0.5 |
| Edit | 39.0/4.7 | 44.3/5.9 | 44.1/6.7 | 46.4/8.8 | 54.6/4.9 | 56.4/6.4 | 3.5/1.1 | 6.2/1.8 |

mately 10.0 on average, although the steps of the two datasets are shared. Besides, Table 3 shows the results on Breakfast and GAIN-B. The performance gap is more obtrusive since the frame accuracy decreases by approximately 60% on average. It indicates that the current methods lack generalizability on the out-of-distribution tasks. We observe that causal-based methods achieve consistent improvements in the OOD scenario. Besides, we find that the performances of LSTM are poor with the F1 score and edit score, while other baseline methods work well. By qualitatively checking the predictions, we find that LSTM only can tell actions from the background, but fails to classify the action categories correctly. It may be because the LSTM model is overly dependent on temporal relations and weakens the representational capacity.

**Quantitative Analysis:** We compare our methods with the baseline methods to demonstrate the effectiveness. Table 2 (the Overall row) and Table 3 summarize the performance comparisons of our methods with all four baseline methods including LSTM(Hochreiter & Schmidhuber, 1997), ED-TCN(Lea et al., 2017), TResNet(He et al., 2016) and MS-TCN++(Li et al., 2020). For all four baseline methods, our causal-based methods achieve significant and consistent improvements on GAIN and obtains comparable results on the original evaluation sets with the frame accuracy metric. For example, after blocking the causal link between $S$ and $Z$, Causal MS-TCN++ relatively outperforms the baseline over +8.1% on GAIN-C. Moreover, on GAIN-B the causal inference methods relatively outperform the baselines over +69.3% on average.

**Domain Analysis:** Following the COIN dataset, we provide more in-depth analysis with experimental results across different domains in Table 2 (Domains are described and showed in the Ap-

pendix). An obvious performance drop from COIN to GAIN-C occurs on domains 'Electrical' and 'Science'. The reason is that steps in these domains often follow a fixed process, which introduces strong contextual dependency to models and results in poor performance on the OOD tasks. The causal inference approach alleviates these negative effects, for example, Causal LSTM relatively outperforms the baseline with a large margin of +9.4% and +10.4% on domain 'Electrical' and 'Science', respectively. On domains like 'Housework', models obtain comparable results on COIN and GAIN-C, which is because the video collection of GAIN-C is independent of the collection for COIN. So it is possible that the videos we find are easier to be segmented than those in COIN. The "Overall" results show that OOD test set is more challenging.

**Qualitative Analysis:** We qualitatively analyze how our method contributes to the improvement of performance. Fig. 5 demonstrates the visualization of two prediction examples on GAIN-C with MS-TCN++ and the corresponding causal method, where different colors means different step categories. Obviously, Causal MS-TCN++ achieves higher frame accuracy on both two examples. At the top, the original MS-TCN++ predicts 6 kinds of steps for the video "Wash cat", while Causal MS-TCN predicts the same causations of steps as the ground truth. This demonstrates that our model does not predict the spurious correlations caused by the context bias but focuses on the step itself. At the bottom, we show an example of the video "Scalded shrimp", the causal one outperforms the baseline method with more smooth predictions.

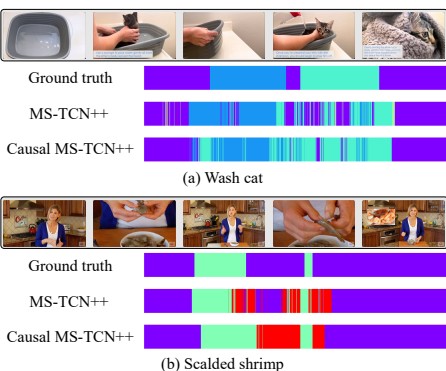

Figure 5: Visualization examples of action segmentation results on GAIN-C.

### 4.3 ACTION DETECTION

**Setting:** The goal of action detection is to detect a series of steps and output the temporal boundaries. It is also an important yet challenging task for IAU. We follow the evaluation protocol of (Lea et al., 2017; Singh et al., 2016) by reporting the widely-used segment-wise metric, mean Average Precision with midpoint hit criterion (mAP@mid). Specifically, the criterion of mAP@mid for a true positive is whether or not the temporal midpoint of the output interval is within the corresponding ground-truth action segments.

**Results:** Table 4 presents the experimental results on COIN, Breakfast, and their counterparts. In this task, we choose LSTM as the baseline. From the training set to GAIN, we observe a huge performance drop by more than 80%, which is related to the weak OOD generalizability of the baseline method. We also compare the causal methods with the baselines. Without any performance cost on the original evaluation set, the causal methods relatively outperform baselines over +33%/+55% on GAIN-C/GAIN-B.

Table 4: Evaluation on training sets and GAIN for action detection. '· / ·' denotes the performances of 'Training set / GAIN'.

| Methods | mAP@mid | |
| --- | --- | --- |
| | COIN | Breakfast |
| LSTM | 32.8 / 6.0 | 57.7 / 8.5 |
| Causal LSTM | 35.2 / 8.0 | 58.6 / 13.2 |

## 5 CONCLUSION

In this paper, we have introduced a dataset, named GAIN, to benchmark the generalizability of IAU models. Our GAIN dataset contains 1,231 videos of 116 OOD tasks, which are collected by reassembling the in-distribution steps of the training set. Based on GAIN, we have proposed to evaluate the generalizability of models with the performance on the OOD tasks. We have also proposed a causal inference approach to cut off the excessive contextual dependency for enhancing generalizability. We evaluate the generalizability of some widely used methods on GAIN and demonstrate that causal inference is a potential direction to improve generalization. We will release this dataset to promote the real-world deployment of IAU models.

**Limitation:** By benchmarking the generalizability on GAIN, we offer a testbed and expect work to develop models that can work in dynamic environments. However, the OOD data needs to be found, which is labor-intensive. Thus, the GAIN dataset cannot be omniscient or cover every aspect. Besides, while causal inference shows its potential to improve generalizability, designing algorithms for generalization is still an open question. It is worth devoting efforts and we leave it as future work.

ACKNOWLEDGEMENT

This work was supported in part by the National Key Research and Development Program of China under Grant 2017YFA0700802, in part by the National Natural Science Foundation of China under Grant 62125603, and in part by a grant from the Beijing Academy of Artificial Intelligence (BAAI).

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

# Appendix

## A    RELATED WORK

**Instructional Action Understanding:** With the explosion of video data on the Internet, learners can acquire knowledge from instructional videos to accomplish different tasks. Many instructional video datasets have been proposed for different goals, such as action detection datasets (Caba Heilbron et al., 2015; Idrees et al., 2017; Gu et al., 2018), video summarization datasets (De Avila et al., 2011; Gygli et al., 2014; Panda et al., 2017; Song et al., 2015), and video caption datasets (Xu et al., 2016; Yu et al., 2018; Miech et al., 2019; Krishna et al., 2017). To analyze instructional videos, diverse research fields are presented in recent years including action segmentation (Richard et al., 2018a;b; Miech et al., 2020; 2019; Sun et al., 2019), procedure segmentation (Zhou et al., 2018b), step localization (Alayrac et al., 2018; Zhukov et al., 2019), action anticipating (Sener & Yao, 2019; Farha et al., 2018), dense video caption (Das et al., 2013a), video grounding (Huang et al., 2017; 2018; Zhou et al., 2018a), and skill determination (Doughty et al., 2018; 2019). Despite the great progress on the in-distribution environment, it is a major challenge to deploy the trained models in the real-world environment.

**Out-of-Distribution Generalization:** How to generalize the trained model into OOD environments is a key challenge in machine learning (Geirhos et al., 2020). A kind of widely-used methods are zero-shot recognition (ZSR) (Xu et al., 2017; Brattoli et al., 2020; Wang et al., 2019) (cross dataset evaluation), where the categories of samples in the testing set are apparently different from the training set. For instructional video, it is difficult to directly recognize unseen step categories. Another evaluation method is Unsupervised Domain Adaptation (UDA) (Busto et al., 2018; Zhang et al., 2019), which trains the model with the data and annotations in the source domain and target domain index (e.g. unannotated target data). Compared with UDA, OOD generalization further considers the setting without target domain information. VDG (Yao et al., 2021) is an OOD generation problem which evaluates the models by the videos with changing the scene or background. However, this setting is more suitable for conventional actions. On the contrast, the key domain changing in

the instructional video is not the scene but the distribution shift of action steps in changing tasks. The detailed comparisons are summarized in Table 1.

**Causal Inference:** Causal inference (Pearl, 2009; 2019) plays an important role in machine learning, which investigates causal effects of different variables. Recently, causal inference has been successfully applied to diverse fields including computer vision (Lopez-Paz et al., 2017; Wang et al., 2020), natural language processing (Park et al., 2019), and reinforcement learning (Nair et al., 2019; Forney et al., 2017), due to its ability for removing confounding bias (Tang et al., 2020; Wang et al., 2020), building explainable machine (Wang & Vasconcelos, 2020; Goyal et al., 2019), promoting fairness (Kusner et al., 2017; Chiappa, 2019), and recovering missing data (Mohan & Pearl, 2018). In this paper, we apply causal inference to mitigate the negative effect brought by confounding context bias to enhance the generalizability of IAU models.

## B PREREQUISITE: CAUSAL MODEL

In this section, we provide some prerequisites of causal model that may help to better understand our causal approach. More details could be found in (Glymour et al., 2016).

Our task for video understanding is to predict the label of step based on the observation as $P(Y|S)$. However, the context steps $Z$ also affect the prediction. With the Bayes rule, we can re-write $P(Y|S)$ as:

$$P(Y|S) = \Sigma_{\boldsymbol{z}} P(Y|S, Z = \boldsymbol{z}) P(Z = \boldsymbol{z}|S), \tag{6}$$

which denotes that the likelihood $P(Y|S)$ are influenced by $P(Z = \boldsymbol{z}|S)$. However, $P(Z = \boldsymbol{z}|S)$ is changed in the OOD setting. Now we use an example to show that $P(Z = \boldsymbol{z}|S)$ introduces the observation bias. In the video "Inflate bicycle tires" , current content $S$ is "installing the nozzle" and the context $Z$ is "using bicycle pump". The content $S$ and the context $Z$ are always observed together in the training process and thus $P(Z = using\ bicycle\ pump|S = installing\ the\ nozzle)$ is higher. It leads the model to predict higher probability $P(Y|S = installing\ the\ nozzle)$ when observing the $Z = using\ bicycle\ pump$ and vice verse. However, when we apply the model to analyze the OOD video "Inflate car tires", where $Z$ "using bicycle pump" is absent, the model may be confused and consequently give wrong prediction. Therefore, we aim at mitigating the influence from $Z$ on $S$. Before that, we first introduce some definitions in the causal inference to help understand, and the proofs can be found in (Pearl, 2009; Glymour et al., 2016).

**Definition 1 (Intervention)** *An intervention represents an external force that fixes a variable to a constant value (akin to random assignment if an experiment), and is denoted $do(S = s)$[1], meaning that $S$ is experimentally fixed to the value $s$.*

**Definition 2 (Confound)** *Consider a pair of variables $S$ and $Y$. Mathematically, $S$ and $Y$ are confounded if*

$$P(Y|S) \neq P(Y|do(S)). \tag{7}$$

**Definition 3 (Confounder)** *$Z$ is a confounder (or $S$ and $Y$ are confounded by $Z$), if $Z$ is associated with $Y$ via paths in the causal graph that are not going through $S$.*

For example, (a) in the causal model $S \leftarrow Z \rightarrow Y$, $Z$ is a confounder because (1) both $S$ and $Y$ are associated with it and (2) it is not on a causal path going through $S$ and $Y$ (3) nor a descendant of them; (b) in the causal model $S \rightarrow Z \rightarrow Y$, $Z$ is not a confounder but a mediator ($Z$ modifies the effect of $S$ on $Y$).

**Definition 4 (The Backdoor Criterion)** *Given an ordered pair of variables $(S, Y)$ in a causal graph, a set of variables $Z$ satisfies the backdoor criterion relative to $(S, Y)$ if (1) no node in $Z$ is a descendant of $S$; (2) $Z$ blocks every path between $S$ and $Y$ that contains an arrow into $S$. If variables $Z$ satisfies the backdoor criterion for $(S, Y)$, then we can adjust the causal effect of $S$ on $Y$ with*

$$P(Y = y|do(S = s)) = \Sigma_z P(Y = y|S = s, Z = z) P(Z = z). \tag{8}$$

---

[1]*do*-operator erases all the arrows that come into $S$ to prevent any information about $S$ from flowing in the non-causal direction.

*In other word, with this criterion we condition on Z such that we (1) block all spurious paths between S and Y; (2) don't disturb any directed paths from S to Y; (3) create no new spurious paths.*

With these definitions, we can show that how the backdoor adjustment can help for our OOD task. When we intervene $S$ with the Back-Door Criterion, i.e. $do(S)$, the link between $S$ and $Z$ is cut off so as the dependency. We formulate the model prediction process under the intervention:

$$P(Y|do(S)) = \Sigma_{\boldsymbol{z}} P(Y|S, Z=\boldsymbol{z})P(Z=\boldsymbol{z}), \tag{9}$$

where $Z = z$ is independent from $S$. The only difference between Eq 6 and Eq 9 is that we change $P(Z|S)$ to $P(Z)$, which shows that $Z$ is no longer affected by $S$. After intervention, when the model predicts from $do(S)$ to the label $Y$, it fairly takes every $z$ into consideration. Thus, the backdoor adjustment can mitigate the negative effect of the biased confounder $Z$.

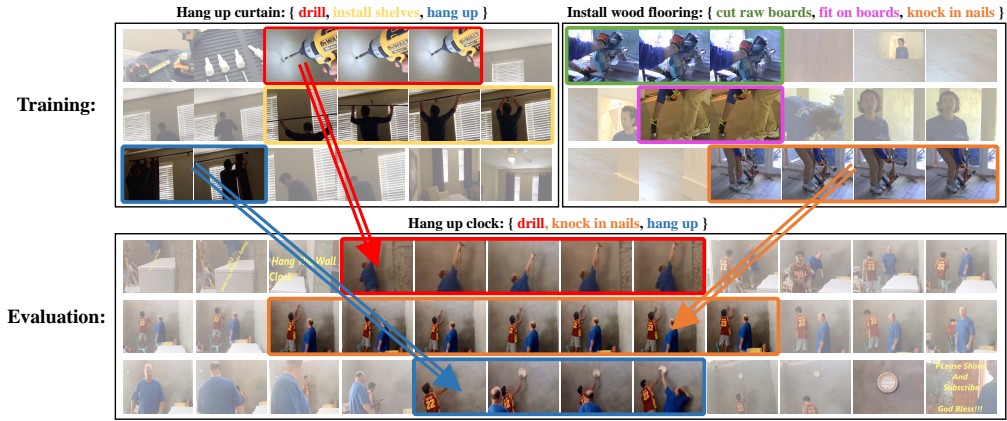

Figure 6: An example of how to generate OOD tasks in GAIN. Given two in-distribution videos of the training sets, we can generate a new OOD task by reassembling the steps of original videos. Best viewed in color.

## C  MORE DETAILS OF GAIN

### C.1  A DETAILED EXAMPLE

To show how to construct the OOD task in GAIN, in Fig. 6, we display a detailed example about "Hang Up Clock" which can be reassembled by the steps in the training tasks "Hang Up Curtain" and "Install Wood flooring". Specifically, the "Hang up curtain" task consists of three steps including $\{drill, install\ shelves,\ hang\ up\}$, and the "Install wood flooring" task is composed of $\{cut\ raw\ boards,\ fit\ on\ boards, knock\ in\ nails\}$. Our collected OOD task "Hang up clock" contains the "drill" and "hang up" steps in the "Hang up curtain" task and the "knock in nails" step in another.

### C.2  TASKS & STEPS

In order to present more details of our GAIN dataset, we show all the selected tasks with their corresponding steps in Table 7, 8, 9 and Table 10. We display these two tables in the end of Appendix because of the typesetting. The GAIN dataset consists of 1,231 instructional videos related to 116 unseen tasks. Each task in GAIN contains 2∼24 videos with an average of 10 videos. We annotate 6382 action segments in GAIN with an average of 5 steps in each video.

### C.3  SAMPLE DISTRIBUTIONS

Fig. 7 and Fig. 8 illustrate the sample distribution of GAIN-C and GAIN-B. Statistically, GAIN-C contains 1,000 videos of 100 unseen tasks with a length of 41.6 hours in total, where 5238 segments are annotated. The GAIN-B split includes 231 videos of 16 OOD tasks with an average length of 2 minutes and 30 seconds. These tasks consist of 20 fine-grained action categories.

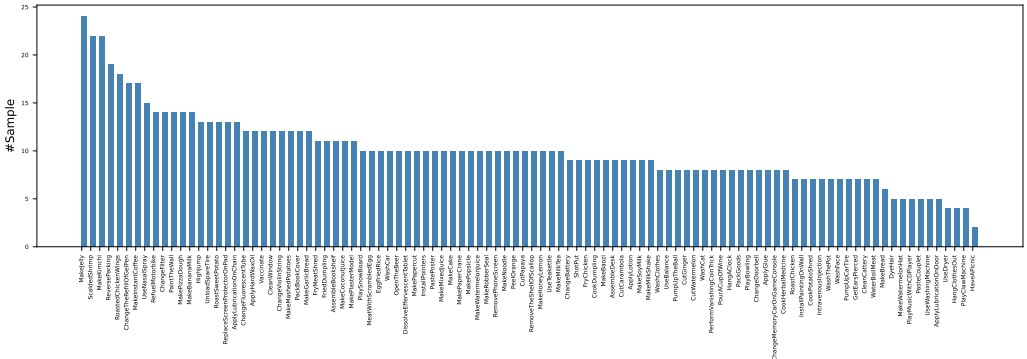

Figure 7: The sample distributions of GAIN-C.

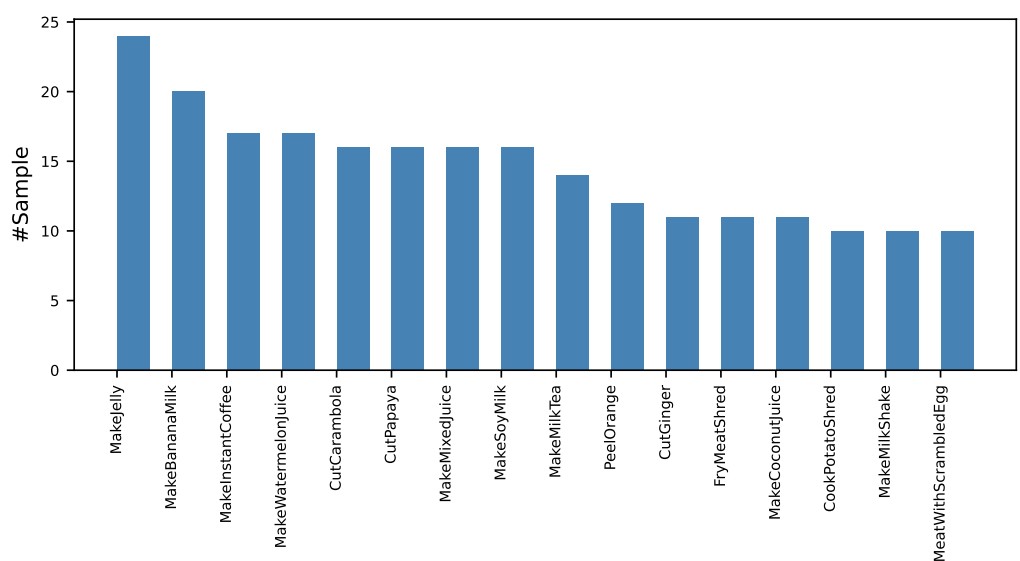

Figure 8: The sample distributions of GAIN-B.

## C.4  DURATION STATISTICS

Fig. 9 illustrates the duration statistics in both video-level and step-level of our GAIN dataset, where the average length of videos is 2 minutes and 30 seconds, and the average length of steps is 12 seconds. Totally, the GAIN dataset contains OOD videos of 51.2 hours for generalizability evaluation.

## C.5  VIEWS ANALYSIS

In Fig. 10, we display the number of views on YouTube across 100 tasks in GAIN-C, which can demonstrate that with the basic principles mentioned in section 3.2, the unseen tasks meet the need of website viewers statistically. We grab the number of views from YouTube by utilizing the Python module $youtubesearchpython$. We form a query with "how to" preceding the task name (e.g. how to paint the wall) to search for YouTube instructional videos related to the tasks. Then for each task, we only extract the first 20 results and sum up the numbers of views to represent the popularity of this task.

With approximately 767.9M views, "Make popsicle" becomes the most-viewed task and the last one "Dissolve effervescent tablet" still obtains 391.8K views. The number of views per task is 44.5M on average and the average number of views for the counted videos is 2.2M. The statistical results above prove that the selected tasks in our GAIN dataset are all common daily tasks and satisfy the

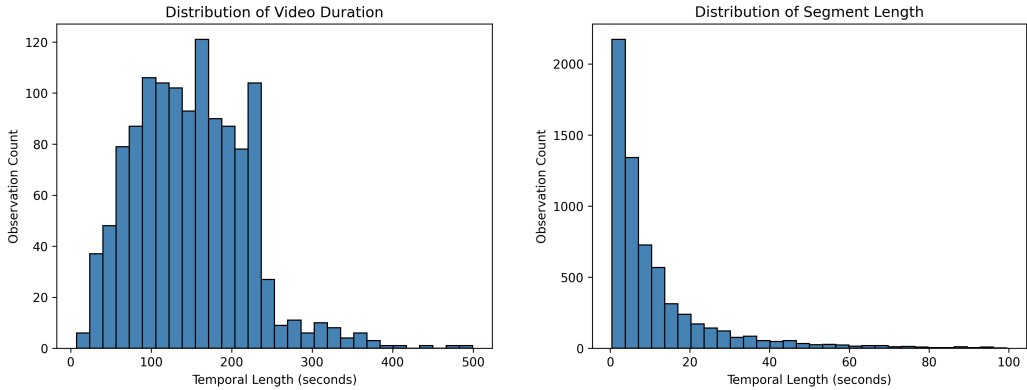

Figure 9: The duration statistics in the video-level (left) and step-level (right) of GAIN.

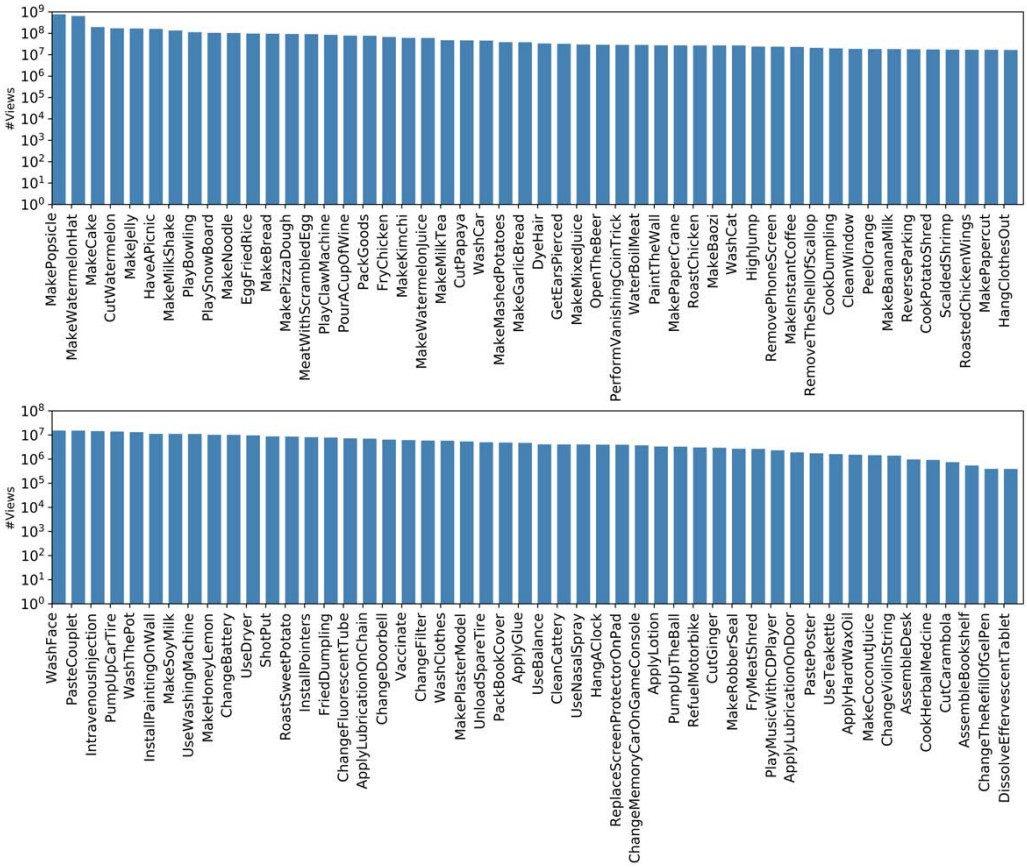

Figure 10: The views distributions of tasks in GAIN-C on YouTube.

demand of website viewers. The learning enthusiasm for diverse tasks reveals the practical value of the generalizability of IAU models.

## C.6 DOMAIN ANALYSIS

Fig. 11 shows the domain distribution (defined in COIN) of this split. GAIN-C covers 12 semantic domains, which demonstrates the rich diversity of our GAIN dataset.

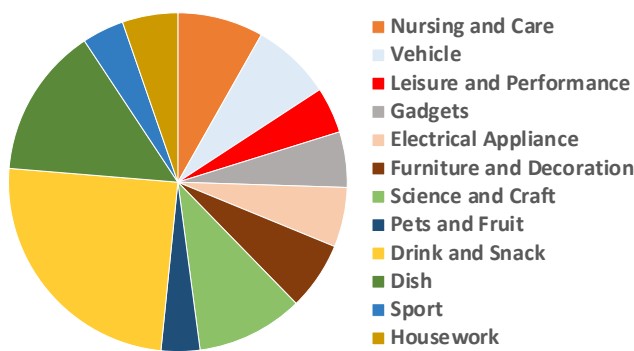

Figure 11: The domain distribution of GAIN-C.

Table 5: Parameter analysis on the learning rate on COIN/GAIN-C.

| Learning Rate | Methods | Frame Accuracy | |
| | | COIN | GAIN-C |
|---|---|---|---|
| 5e-4 | MS-TCN++ | 62.1 | 49.0 |
| | Causal MS-TCN++ | **64.0** | **52.3** |
| 1e-3 | MS-TCN++ | 64.7 | 54.3 |
| | Causal MS-TCN++ | **65.5** | **56.2** |
| 2e-3 | MS-TCN++ | 65.5 | 51.8 |
| | Causal MS-TCN++ | **65.6** | **56.0** |

Table 6: Parameter analysis on different relative sizes of reassembled videos on Breakfast/GAIN-B with frame accuracy.

| Method | EDTCN | Causal ED-TCN | | | | |
|---|---|---|---|---|---|---|
| Step # | - | 1.x | 0.5x | 1.5x | 1.x | 1.x |
| Video # | - | 1.x | 1.x | 1.x | 0.5x | 1.5x |
| Accuracy | 42.4/21.3 | 44.5/35.4 | 44.5/31.1 | 44.5/30.7 | 43.9/29.1 | 42.1/37.6 |

# D  MORE EXPERIMENTAL ANALYSIS

## D.1  PARAMETER ANALYSIS

We conduct experimental analysis on both COIN/GAIN-C to investigate the effect of hyper-parameter learning rate with MS-TCN++(Li et al., 2020). As shown in Table5, with a learning rate of $5e-4$ the model performs unfavorable results on both two datasets, while an increased learning rate, i.e. $1e-3$ or $2e-3$, can notably improve its performance on COIN($+2.3\%$ and $+2.5\%$) as well as the performance on the out-of-distribution tasks ($+7.5\%$ and $+7.1\%$).

We also conduct experiments of different relative sizes of reassembled videos on Breakfast/GAIN-B and the results (frame accuracy) are shown in Table 6. For the number of steps, $1.0\times$, $0.5\times$, and $1.5\times$ of Causal ED-TCN denote that we use 1 times, 0.5 times, and 1.5 times step numbers as that of the original ED-TCN. So as the number of videos. Our methods with different setting consistently outperform the baseline on both in-distribution and OOD scenario. Besides, we find that with the same number of augmented steps and videos the model achieves better overall performance, so we adapt this setting for all experiments.

## D.2  VISUALIZATION RESULTS

In section 5.2, we have visualized the ground-truth annotations and the action segmentation results. Due to the limitation of space, we only visualized the results produced by MS-TCN++ method and its causal version. Now in Fig. 12, we illustrate the visualization for different methods including LSTM, ED-TCN, TResNet and MS-TCN++, to demonstrate the effectiveness of our approach.

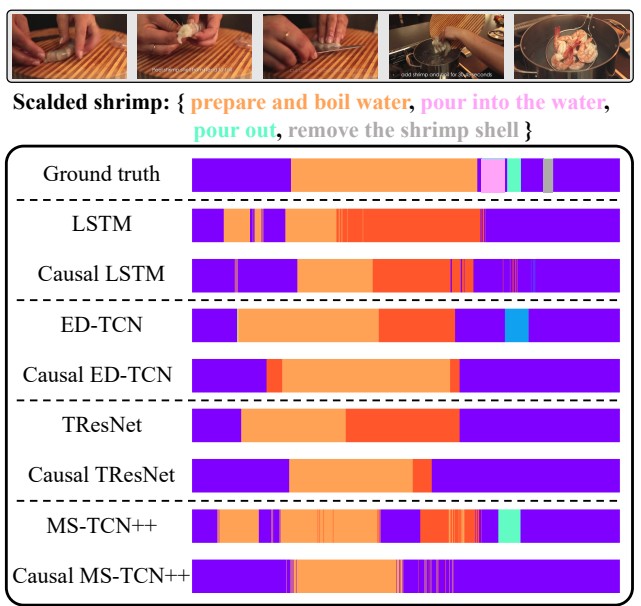

Figure 12: Visualization of action segmentation results. The video is associated with the task "Scalded shrimp". Best viewed in color.

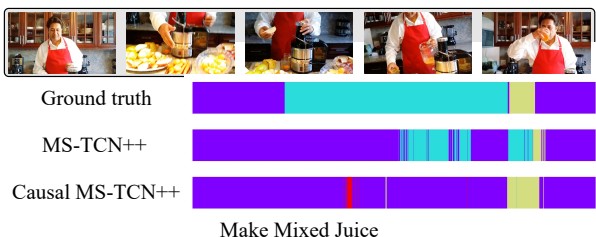

Make Mixed Juice

Figure 13: Visualization of a failure case. The video is associated with the task "Make Mixed Juice". Best viewed in color.

We show results of the baseline methods and our corresponding causal ones on the task "Scalded shrimp" in GAIN-C. The consistent improvements for different baselines on out-of-distribution tasks indicate that our causal intervention promotes the generalizability of models.

Additionally, we show a failure case of our approaches in Fig. 13. We analyze the underlying insights from the cases that the causal-based method has lower performance than the baseline, such as the video "Make Mixed Juice" in the GAIN-C dataset. Take a closer look, we find that there are some strong step dependencies, such as "juice the fruit" and "pour the juice". In this situation, context bias has positive effects on the evaluation. Thus, we got the following insights, despite the better performance on average, our method encourages step independency, which has negative effects on the examples where strong step dependencies occur for both training and testing data. It is reasonable since these steps are in-distribution samples where the context bias has a positive effect, e.g., "juice the fruit" and "pour the juice" are successive steps in both COIN and GAIN-C, so the prediction with the concurrence of them is better.

Table 7: Tasks and the corresponding steps in GAIN-C.

| Tasks | Steps | Tasks | Steps |
|---|---|---|---|
| ApplyGlue | apply glue to the wall and wallpaper, paste and level the wallpaper | MakeMixedJuice | cut ingredients, juice the oranges, pour the orange juice into the cup |
| ApplyHardWaxOil | apply the cleaning agent with towel evenly, wipe off the cleaning agent | MakeNoodles | add some water to the tea, mix the raw materials, knead the dough, cut the flesh |
| ApplyLotion | pour some glue to the face, wipe the glue to a layer | MakePaperCrane | fold or bent paper, paint on the paper |
| ApplyLubrication OnChain | apply the lubricant on the lock, wipe off the redundant lubricant, check the old chain | MakePapercut | draw an outline, cut along the edges, fold or bent paper |
| ApplyLubrication OnDoor | apply the lubricant on the lock, wipe off the redundant lubricant | MakePizzaDough | mix the raw materials, add some water to the tea, rub and drag the materials, cut the flesh |
| AssembleBookshelf | assemble the frame, install horizontal boards, install vertical boards | MakePlasterModel | mix the raw materials, soak them in water, wait for the candle until concretion, remove the gill, add raw materials, add some water to the tea |
| AssembleDesk | install stand of the seat, install legs on the bed | MakePopsicle | pour the ingredients into the bowl, put the candle wick into a vessel, put the melted soap block into the vessel, take out after freezing, mix the raw materials |
| ChangeBattery | open the back cover, replace the battery, install the back cover and waterproof ring | MakeRobberSeal | draw an outline, carve along the outline, remove the peel |
| ChangeDoorbell | screw off the screws used to fix the switch, reset the switch and screw on the screws used to fix the switch | MakeSoyMilk | soak and wash the rice, put yogurt, honey and other ingredients into the juicer, add some water to the tea, mix raw materials, pour the tea into the vessel, juice the oranges |
| ChangeFilter | take out the old filter, remove the cap of the new filter, install the new filter, rinse the dish | MakeWatermelonHat | clean up the interior of thepumpkin, carve along the outline, draw an out line |
| ChangeFluorescentTube | take out the old bulb, install the new bulb, remove the light shell/housing/support, install the light shell/housing/support | MakeWatermelonJuice | cut oranges, juice the oranges, pour the orange juice into the cup |
| ChangeMemoryCard OnGameConsole | use the needle to open the SIM card slot, put the SIM card into the SIM card slot, press the SIM card slot back | MeatWithScrambledEgg | pour the egg into the bowl, stir the egg, prepare meat, cut the flesh, mix the raw materials, put in the oil to fry |
| ChangeTheRefillOf GelPen | remove cap, put lead into the pen, buckle the cap | OpenTheBeer | open the bottle carefully |
| ChangeViolinString | cut off and remove the old string, fix the new string on the lower part of the guitar, fix the new string on the head of the guitar, adjust the tightness of the new string | PackBookCover | measure the size of the packing paper, cut the packing paper, fold or bent paper |
| CleanCattery | remove the toy and paper bed from the hamster cage, clean toys and hamster cages, move the toy and paper bed into the hamster cage | PackGoods | remove cap, put in the battery, close cover |
| CleanWindow | apply the cleaning agent with towel evenly, wipe off the cleaning agent | PaintTheWall | dip the glue, apply glue to the wall and wallpaper |
| CookDumpling | knead the dough, flatten the dough, add ingredients into cone, knead together, mix raw materials, soak them in water, load the dish | PasteCouplet | apply glue to the wall and wallpaper, paste and level the wallpaper |
| CookHerbalMedcine | prepare and boil water, pour the noodles into the water and stir, filtrate with a filter, pour into a glass | PastePoster | apply glue to the wall and wallpaper, paste and level the wallpaper |
| CookPotatoShred | peel, cut into strips and pieces, put in the oil to fry | PeelOrange | remove the peel, cut the flesh |
| CutCarambola | cut off the edge, cut the flesh | PerformVanishing CoinTrick | show the glass to the audience, block out the glass, show the vanished glass |
| CutGinger | remove the peel, slice the pulp | PlayBowling | pre-swing, push curling |
| CutPapaya | peel, cut in half, dig out the seeds with spoon, slice the pulp | PlayClawMachine | insert money into the vending machine, press the corresponding button, take out the goods |
| CutWatermelon | cut off the edge, peel, cut in half, slice the pulp | PlayMusicWith CDPlayer | take out the laptop CD drive, press the SIM card slot back, close the fuel tank cap |
| DissolveEffervescent Tablet | add some ingredients to the tea, add some water to the vessel | PlaySnowBoard | ski down from the hill, ski up from the hill, rise to the sky |

Table 8: Tasks and the corresponding steps in GAIN-C.

| Tasks | Steps | Tasks | Steps |
|---|---|---|---|
| DyeHair | apply the shampoo or hair conditioner, scratch the hair carefully, wash the body wash away, make the hair dry | PourACupOfWine | open the bottle carefully, pour in after mix it |
| EggFriedRice | take out some rice, soak and wash the rice, pour the egg into the bowl, stir the egg, mix raw materials, put in the oil to fry | PumpUpCarTire | screw off the valve cap and open the valve, install the air nozzle, remove the air nozzle, tighten the valve and screw on the valve cap |
| FriedDumpling | knead the dough, flatten the dough, cut the flesh, add ingredients into cone, knead together, put in the oil to fry, mix the raw materials, add some water to the tea | PumpUpTheBall | install the air nozzle, pump up to the tire, remove the air nozzle |
| FryChicken | prepare seasonings and side dishes, prepare meat, cut the flesh, fry or gril | RefuelMotorbike | open the fuel tank cap, insert oil gun in the car, pullthe oil gun out, close the fuel tank cap |
| FryMeatShred | prepare meat, cut the flesh, put in the oil to fry, load the dish | RemovePhoneScreen | unscrew the screws used to fix the screen, pull out the screen connector and remove the screen |
| GetEarsPierced | draw lines to mark the hole, find the position of the hole, drill with an electric drill | RemoveTheShellOf Scallop | cut oranges, take out the shell, rinse the pot, remove the gill, open up the cover |
| HangAClock | drill in the wall, knock in the nails, hang up curtains | ReplaceScreenProtector OnPad | wipe the screen, paste protector on the screen |
| HangClothesOut | wrap the hair by hands, hang the ironed clothes | ReverseParking | drive the car forward, drive the car backward, adjust front and back position |
| HaveAPicnic | clean up the ground, lay the cushion evenly, load the dish | RoastChicken | prepare seasonings and side dishes, remove the intestines and blood vessels, brush sauce or sprinkle seasoning, bake pizza |
| HighJump | begin to run up, begin to jump up, fall to the ground | RoastedChickenWings | soak them in water, add raw materials, mix the raw materials, bake pizza |
| InstallPaintingOnWall | drill in the wall, knock in the nails, paste and level the wallpaper | RoastSweetPotato | clean the pumpkin, fry or roast or braise, cut the bread, peel |
| InstallPointers | let the flat side of the new needle towards the jack and insert the new needle, screw on the screw | ScaldedShrimp | prepare and boil water, pour the noodles into the water and stir, remove the shrimp shell, pour the cooked noodles |
| IntravenousInjection | tie the tourniquet, disinfect, pull out the needle and press with cotton | ShotPut | pre-swing, throw the hammer out |
| MakeBananaMilk | peel, cut into strips and pieces, add milk, shake and juice, pour the orange juice into the cup, put strawberries and other fruits into the juicer | UnloadSpareTire | unscrew the screw, remove the tire |
| MakeBaozi | add ingredients into cone, knead together | UseBalance | put the sample to be measured on the balance, put the weight until the balance is balanced |
| MakeBread | knead the dough, run the toaster and adjust, take out the slice of bread | UseDryer | put the clothes neatly on a ironing table, use a hair dryer to blow hot wall, flip the clothes repeatly |
| MakeCake | pour the egg into the bowl, add raw materials, mix raw materials, put materials into mold, run the toaster and adjust, take out chocolate | UseNasalSpray | wipe nose, fill a nostril with saline and do the same to the other nostril, shake and stir, remove cap |
| MakeCoconutJuice | dig out the seeds with spoon, put the ingredients into the can, pour the orange juice into the cup, put strawberries and other fruits into the juicer, put yogurt, honey and other ingredients into the juicer, shake and juice | UseTeakettle | pour the tea into the vessel, heat the teapot and wash the cup, close up the cover |
| MakeGarlicBread | mix the raw materials, cut the bread, put the filler on the bread slice, put a slice of bread in, run the toaster and adjust, take out the slice of bread | UseWashingMachine | open the fuel tank cap, add some cleaner to clean and wet the lenses and take out the lenses, close up the cover |
| MakeHoneyLemon | cut ingredients, add some water to the tea, put the ingredients into the can, mix and pickle | Vaccinate | fill the injection head, disinfect the injecting place, inject to the muscular, pull out the needle and press |
| MakeInstantCoffee | add tea powder, brew tea and stir, add some ingredients in the coffee, mix the raw materials | WashCar | add detergent and make bubble, clean the surface, wipe off the cleaning agent |
| MakeJelly | pour the ingredients into the bowl, stir the mixture, take out chocolate, cut into strips and pieces, put materials into mold | WashCat | use the body wash, wash the body wash away |

Table 9: Tasks and the corresponding steps in GAIN-C.

| Tasks | Steps | Tasks | Steps |
|---|---|---|---|
| MakeKimchi | cut into strips and pieces, mix and pickle, clean up and soak, put the ingredients into the can, mix raw materials, add some water to the tea, shake and juice | WashClothes | add detergent and make bubble, clean toys and hamster cages, wrap the hair by hands |
| MakeMashedPotatoes | peel, cut potato into strips, soak them in water, add raw materials, mix raw materials, prepare and boil water | WashFace | wet and wash hands, apply cleansing milk to the face, wipe up the face |
| MakeMilkShake | add milk, shake and juice, pour the orange juice into the cup | WashThePot | add detergent and make bubble, flush and wash the interior, scrub the toilet interior |
| MakeMilkTea | put yogurt, honey and other ingredients into the juicer, mix the raw materials, pour in after mix it, add ice cubes | WaterBoilMeat | prepare and boil water, soak them in water, load the dish |

Table 10: Tasks and the corresponding steps in GAIN-B.

| Tasks | Steps |
|---|---|
| CookPotatoShred | peel fruit, cut orange, pour oil |
| CutCarambola | cut orange |
| CutGinger | peel fruit, cut fruit |
| CutPapaya | peel fruit, cut fruit |
| FryMeatShred | cut orange, pour oil, take plate |
| MakeBananaMilk | peel fruit, cut orange, pour milk, squeeze orange, pour juice, put fruit to bowl |
| MakeCoconutJuice | put fruit to bowl, pour juice, squeeze orange |
| MakeInstantCoffee | add teabag, stir coffee |
| MakeJelly | pour flour, stir dough, cut bun, put fruit to bowl |
| MakeMilkShake | pour milk, squeeze orange, pour juice |
| MakeMilkTea | pour milk, stir tea, pour water, put fruit to bowl |
| MakeMixedJuice | cut orange, squeeze orange, pour juice |
| MakeSoyMilk | stir fruit, pour milk, pour water, stir tea, squeeze orange |
| MakeWatermelonJuice | cut orange, squeeze orange, pour juice |
| MeatWithScrambledEgg | pour egg into pan, stir fry egg, cut orange, stir fruit, pour oil |
| PeelOrange | peel fruit, cut orange |

