# OpenReview forum: "GAIN: On the Generalization of Instructional Action Understanding"
_ICLR.cc/2023/Conference — ICLR 2023 poster_

### Official Review · Reviewer_2UkL · 2022-10-23

**Confidence:** 4
**Correctness:** 4
**Technical Novelty And Significance:** 3
**Empirical Novelty And Significance:** 4
**Recommendation:** 8

**Clarity, Quality, Novelty And Reproducibility:**

There is no reproducibility issue as codes provided. This paper propose a novel OOD evaluation for instructional action understanding.

**Strength And Weaknesses:**

Strength:
* Introduce an interesting OOD evaluation definition: known steps but unknown tasks. As same steps have appearance bias in different tasks, it is suitable for generalization ability evaluation.
* Introduce a reasonable pipeline for constructing dataset.
* Propose an effective module to improve generalization ability.

Weekness:
* I suggest to simply introduce the definition of action segmentation / action detection tasks in Section 2 rather than Section 4. For readers who are not familiar with these two tasks, it is confusing to read Section 2. Meanwhile, I also suggest to simply introduce how a SOTA action segmentation method (such as TCN) works on Section 1 or 2. These background knowledge can help readers better understand the proposed task and method.
* The dataset size still seems a little small for so many unseen tasks.
* For annotation of collected  videos, I am still confusing how un-defined steps are annotated. "Furthermore, explicit steps in a video do not need to exactly match those in its task – in other words, permuting and being a proper subset of the task are acceptable. " If a collected video contains un-defiend step, will this step be annotated? If it is annotated, will it be considered during evaluation? This is an important point and authors should give clear description.

**Summary Of The Paper:**

This paper introduce a new benchmark for evaluating the generalization ability of instructional action understanding models. The main idea is to collect existing steps in training dataset and re-assembling them as new task, then collect corresponding videos. Authors also propose an effective casual-based  method to improve generability.

**Summary Of The Review:**

In conclusion, this work proposes a reasonable OOD evaluation task for instructional action understanding, along with dataset and baseline method. My concern is mainly about the organization of the paper content as aforementioned. I give "accept, good paper" as my rating.

---

> ### Author Response · Authors · 2022-11-16
> **Response to Reviewer 2UkL**
>
> Dear Reviewer 2UkL, we are sincerely grateful for your careful reading and valuable feedback, which has helped improve our paper. We provide the point-to-point response below and have updated the paper and appendix accordingly.
>
> > **Q1:** I suggest to simply introduce the definition of action segmentation / action detection tasks in Section 2 rather than Section 4. For readers who are not familiar with these two tasks, it is confusing to read Section 2. Meanwhile, I also suggest to simply introduce how a SOTA action segmentation method (such as TCN) works on Section 1 or 2. These background knowledge can help readers better understand the proposed task and method.
>
> **A1:** Thanks very much for your kind suggestion! In light of your good suggestion, we've given an brief introduction for both action segmentation and action detection tasks in Section 1, and added an introduction about TCN in to make it more readable. Please kindly refer to our revised paper (the first paragraph and the paragraph above Contributions in Section 1) for details.
>
> > **Q2:** The dataset size still seems a little small for so many unseen tasks.
>
> **A2:** Thanks for pointing out this. We plan to collect more instructional video data and update the dataset annually for several years, like some video datasets (Kinetics, YouTubeVOS, etc). It is not trivial since the data collection, filtering, and annotation are labor intensive. Besides, the current size of GAIN is comparable to the test sets of the original datasets, i.e. COIN and Breakfast, which is adequate for preliminary evaluation. We expect the introduction of GAIN will abstract attention to the generalization problem of instructional video understanding tasks.
>
> > **Q3:** For annotation of collected videos, I am still confusing how un-defined steps are annotated. "Furthermore, explicit steps in a video do not need to exactly match those in its task – in other words, permuting and being a proper subset of the task are acceptable. " If a collected video contains un-defiend step, will this step be annotated? If it is annotated, will it be considered during evaluation? This is an important point and authors should give clear description.
>
> **A3:** Thanks for your good question. The undefined steps will be considered as the background, which does not need to be further annotated. On the one hand, during the data collection stage, if a video contains a long stretch of undefined but important steps, this video will be filtered out according to the principle of Step Consistent. Taking the unseen task “Assemble a Desk” as an example, boards of the desk can be bonded with screws or glues, so both "screw" and "apply glues" are important steps. If "screw" is a defined step and "apply glues" is undefined, the videos in which glues are used will be filtered out. On the other hand, videos with undefined meaningless steps are acceptable and these steps won't be annotated. For example, if "wipe the board" appears in a video of "Assemble a Desk", this video is acceptable and this step will be considered as the background. We've given a clear description of how to deal with undefined steps in the revised version (Please refer to Sec. 2.3).

---

### Official Review · Reviewer_ESEp · 2022-10-23

**Confidence:** 3
**Correctness:** 3
**Technical Novelty And Significance:** 4
**Empirical Novelty And Significance:** 4
**Recommendation:** 6

**Clarity, Quality, Novelty And Reproducibility:**

- Clarity: some descriptions in the paper are currently not clear enough.
- Quality: the quality of the paper is generally ok.
- Novelty: the contributions of the paper are novel.
- Reproducibility: Some additional details are required for one to reproduce the results in the paper, e.g., how to determine the size of the augmented data. In addition, the authors provided the code but I did not check.


**Strength And Weaknesses:**

Strength:
1. The OOD evaluation sets (i.e., the GAIN benchmark) are novel and valuable to the research community.
2. According to the experimental results, the proposed causal-inference-based method seems to be able to enhance generalizability of models if the instruction video understanding task is a step-level task.

Weaknesses:
1. Some assumptions need to be stated. For example, the proposed method assumes the constructed casual graph shown in Figure 4(a) is correct, so the proposed method assumes the step S would not affect the context steps Z (but one may argue this is not necessarily to be true in reality).
2. The proposed causal-inference-based method is effective only when the instructional video understanding task is focusing on the steps. I don’t think the proposed method would be beneficial if the instructional video understanding task is a more coarse-grained task such as video task recognition, domain recognition, etc., or tasks that require global context consistency, since the ressembed/intervened videos are not necessarily to be temporally semantically reasonable (because irrelevant steps are combined together to form an intervened video).
3. CrossTask might be the most similar dataset to GAIN, but the paper only briefly describes the difference between CrossTask and GAIN in the introduction. Some statistics to support that the shared steps are only a minority in CrossTask but this is not true for GAIN, would be great to have.
4. Some descriptions are not clear and additional explanations are needed.
- The causal-inference-based method is technically a data augmentation method, then, how would the relative size of the augmented data (relative compared to the original data) affect the performance? What is the current size of the augmented data? Since the ressembed/intervened videos are not necessarily to be temporally semantically reasonable, I assume the relative size of the augmented data has to be small in order to be effective.
- Why temporal boundaries of all steps of the training data is a prerequisite for finding unseen tasks (last sentence of the first paragraph of Sec. 2.2)? To my understanding, one would only need the set of steps in training, and use these steps to form novel combinations to obtain the novel task candidates, which are further filtered and collected via human annotations.
- “... the larger number of categories is better” (from the Category Diverse paragraph). What kind of categories? For a given novel task, the collected videos of that task are supposed to be diverse. However, it is still not clear to me how the diversity is defined.
- “... apply the steps in the training dataset as the anchor steps and generate 10 step combinations with the caption clues”. What do you mean by “with caption clues”?
5. Failure case analysis of the proposed method is currently missing.
6. Limitations of the proposed benchmark and the proposed method were not discussed.

Other comments:
1. In Figure 3 (a), there are some prominent spikes. It is interesting to know what are these top frequent steps in COIN Train, COIN Test, and GAIN-C.
2. Would the causal based method reduce the number of unique steps a video has in predictions (compared to before applying the causal-based method)?


**Summary Of The Paper:**

Summary:
- The paper proposed out-of-distribution (OOD) evaluation sets for the COIN and Breakfast dataset, together referred to as the new GAIN benchmark. To formulate the novel OOD evaluation sets, the authors ensured that the training set and the OOD evaluation set share the same set of steps, but different sets of tasks.
- Empirical results were shown that the OOD evaluation sets are indeed more challenging as they require advanced model generalizability.
- A causal-inference-based method, operated as data augmentation, is further proposed that allows the existing models to enhance generalizability and also perform well on the OOD evaluation sets.


**Summary Of The Review:**

The paper proposed the GAIN benchmark, out-of-distribution evaluation sets for step-level understanding tasks for instructional videos. A causal-inference-based method, operated as data augmentation, is further proposed that allows the existing models to enhance generalizability. The proposed GAIN benchmark would be valuable for the instructional-video-understanding community.

However, clarity of the paper needs to be improved. Some assumptions, explanations, and implementation details were not clearly described. Though the proposed novel components of the paper are interesting, the paper needs more in-depth discussions.

---

> ### Author Response · Authors · 2022-11-16
> **Response to Reviewer ESEp, Part I**
>
> Dear Reviewer ESEp, we sincerely thank you for the time dedicated to reviewing this paper and the helpful comments. We made the revisions according to your suggestions. Please see the revised paper and appendix for details. Below we give a point-by-point response to the comments.
>
> > **Q1:** Some assumptions need to be stated. For example, the proposed method assumes the constructed casual graph shown in Figure 4(a) is correct, so the proposed method assumes the step S would not affect the context steps Z (but one may argue this is not necessarily to be true in reality).
>
> **A1:** Thanks for your good suggestions! Firstly, the casual graph we constructed in Figure 4(a) describes the information flow during the inference. When S is being estimated, other context steps are Z, and since S is affected by Z during the inference, Z points to S in the causal graph (Please refer the second paragraph that describes the causal graph). Secondly, although there is actually bidirectional effect between S and Z, we find that Bayes' theorem remains unchanged, i.e., $P(S,Z)= P(S)P(Z|S)$ and Eq.(3) can still be written as $P(Y|S)= \sum_{z} P(Y|S,Z=z)P(Z=z|S)$. The proposed method aims to cut off the dependency between S and Z, so both unidirectional or bidirectional effect would be cut off. Furthermore, because the causal effect $S \rightarrow Z \rightarrow Y$ won't introduce bias and only $Z\rightarrow S$ will, the proposed method analyzes the task under this circumstance. We only illustrate $Z\rightarrow S$ to highlight that our method is addressing the bias brought by confounder $S \leftarrow Z \rightarrow Y$, which is validated in Figure 3. We've revised the paper and provided explanation (Please refer to the paragraph below Eq.(3)). If the illustration of bidirectional effect is necessary from your perspective, we would like to revise the causal graph in our final version.
>
> > **Q2:** The proposed causal-inference-based method is effective only when the instructional video understanding task is focusing on the steps. I don’t think the proposed method would be beneficial if the instructional video understanding task is a more coarse-grained task such as video task recognition, domain recognition, etc., or tasks that require global context consistency, since the ressembed/intervened videos are not necessarily to be temporally semantically reasonable (because irrelevant steps are combined together to form an intervened video).
>
> **A2:** Thanks very much for pointing it out. The proposed method is only focusing on the steps when understanding  instructional videos and will reach its boundary with regard to more coarse-grained tasks. The paper investigates the OOD generalization of instructional action whose key obstacle is the distribution shift of action steps under different task categories (shown in Figure 3), so it focuses more on steps. We want to highlight our step-based OOD task and task-level video domain generalization (VDG) are two different tasks. In the related work in the appendix, we briefly reviewed VDG, which focuses on the changes of scene and background.  The methods for task-level VDG rely on the invariant temporal representation, which is not suitable for our step-based OOD task.
> Besides, investigating the step-based OOD problem is more suitable for real-world instructional video applications, because users are more curious about the fine-grained steps than the tasks. Given learning from the video of "Roast Chicken", users are more likely to know how to roast different food such as duck instead of how to accomplish this task in different kitchens.
>
> > **Q3(1):** CrossTask might be the most similar dataset to GAIN, but the paper only briefly describes the difference between CrossTask and GAIN in the introduction.
>
> **A3(1):** Thanks for your suggestion. There are two notable differences between CrossTask and GAIN. First, CrossTask is built to investigate whether sharing constituent components improves the performance of weakly supervised learning. In contrast, GAIN is used to analyze generalization and our focus is OOD steps instead of components. Second, the related tasks in CrossTask are not fine-grained annotated, which cannot be used for evaluation in our setting. Besides, only partial steps are shared among tasks in CrossTask, which is not qualified according to the principle of Step Consistent. We've revised the introduction and provided more comparisons (Please refer to the third paragraph of the introduction).
>
> > **Q3(2):** Some statistics to support that the shared steps are only a minority in CrossTask but this is not true for GAIN, would be great to have.
>
> **A3(2):** Thanks for this valuable suggestion to improve our readability. We've counted the steps in CrossTask and found that only 14% of steps are shared among tasks (73 are shared of a total of 517 steps). Moreover, all steps in GAIN are shared with the original training set. We've added statistics to support the argument.

---

> ### Author Response · Authors · 2022-11-16
> **Response to Reviewer ESEp, Part II**
>
> > **Q4(1):** The causal-inference-based method is technically a data augmentation method, then, how would the relative size of the augmented data (relative compared to the original data) affect the performance? What is the current size of the augmented data? Since the ressembed/intervened videos are not necessarily to be temporally semantically reasonable, I assume the relative size of the augmented data has to be small in order to be effective.
>
> **A4(1):** Thanks very much for this suggestion.  In light of your suggestions, we added a parameter analysis experiment for the relative size of the augmented data in the appendix of the revised paper (Please refer to the second paragraph of Sec. C.1).  The current number of sampled steps in one reassembled video is the same as the original video, and the current numbers of reassembled videos and original videos are also the same. As shown in the table below, we summarized the results (frame accuracy) on Breakfast / GAIN-B for different relative sizes of the augmented data. In the table, for the number of steps, $1.0\times $, $0.5\times$, and $1.5\times$ of Causal ED-TCN denote that we use 1 time, 0.5 times, and 1.5 times step numbers as that of the original ED-TCN. So as the number of videos. Our methods with different settings consistently outperform the baseline on both in-distribution and OOD scenarios. Besides, we find that with the same number of augmented steps and videos the model achieves better overall performance, so we adapt this setting for all experiments.
>
> Table A: Results of different relative sizes of the augmented data on Breakfast/GAIN-B with frame accuracy.
> | Method   |   EDTCN   |           |           | Causal ED-TCN |           |           |
> |----------|:---------:|:---------:|:---------:|:-------------:|:---------:|:---------:|
> | Step #   |     -     |    1.x    |    0.5x   |      1.5x     |    1.x    |    1.x    |
> | Video #  |     -     |    1.x    |    1.x    |      1.x      |    0.5x   |    1.5x   |
> | Accuracy | 42.4/21.3 | 44.5/35.4 | 44.5/31.1 |   44.5/30.7   | 43.9/29.1 | 42.1/37.6 |
>
> > **Q4(2):** Why temporal boundaries of all steps of the training data is a prerequisite for finding unseen tasks (last sentence of the first paragraph of Sec. 2.2)? To my understanding, one would only need the set of steps in training, and use these steps to form novel combinations to obtain the novel task candidates, which are further filtered and collected via human annotations.
>
> **A4(2):** Thanks for pointing out this. Here, we use "prerequisite" to show that we could refer to the video content for details with the temporal boundary of steps so that we can understand the steps and annotate unseen task candidates better. We remove this confusing description in the revised paper. If it is necessary from your perspective to show this information, we will find a way to add this sentence as: "Since fine-grained annotations in both datasets provide the temporal boundaries of all steps, these two datasets could provide details for steps with explicit video content, which helps to understand action steps and find unseen tasks."
>
> > **Q4(3):** “... the larger number of categories is better” (from the Category Diverse paragraph). What kind of categories? For a given novel task, the collected videos of that task are supposed to be diverse. However, it is still not clear to me how the diversity is defined.
>
> **A4(3):** Thanks for pointing it out to reduce the confusion. In the Category Diverse paragraph, the diversity is related to the number of unseen task categories. This paragraph introduced one of the principles for finding unseen tasks, so "diverse videos" is used to describe task categories. We argue that the larger number of task categories is the better. For example, a dataset (with 3 tasks) contains 2 videos of repairing a car, 2 videos of repairing a roof and a video of repairing a television is more diverse than a dataset (with only 1 task) with 5 videos of repairing a car. We've revised this paragraph to eliminate ambiguity (Please refer to the Category Diverse paragraph in Sec. 2.2).

---

> ### Author Response · Authors · 2022-11-16
> **Response to Reviewer ESEp, Part III**
>
> > **Q4(4):** “... apply the steps in the training dataset as the anchor steps and generate 10 step combinations with the caption clues”. What do you mean by “with caption clues”?
>
> **A4(4):** There are usually many captions in instructional videos and they could help to generate step combinations. For example, in a video of installing wood flooring, there is a caption, "If you want to fasten the flooring on the wall, you firstly need to drill in the wall", including "drill in the wall" which are not a step in current video. However, "drill in the wall" is closely related to other steps in the video, like "knock in the nails", so "drill in the wall" together with "knock in the nails" is more likely to compose part of a reasonable combination, which finally helps to find an unseen task Install Painting On Wall. We've revised the paper and provided explanation (Please refer to the paragraph below Category Diverse in our revised manuscript).
>
> > **Q5:** Failure case analysis of the proposed method is currently missing.
>
> **A5:** Thanks for your kind suggestion! We've added failure case analysis with corresponding visualization in Appendix (Please refer to the second paragraph of Sec. C.2). We observe that both the baseline and the causal method achieve poor performance in some cases, even though the causal method performs better than the baseline.
>
> > **Q6:** Limitations of the proposed benchmark and the proposed method were not discussed.
>
> **A6:** Thanks for your suggestion and we've discussed limitations in the revised version (Please refer to Sec. D). By benchmarking the generalizability on GAIN, we offer a testbed and expect work to develop models that can learn transferable knowledge and work in dynamic environments. However, the OOD data needs to be found, which is labor-intensive. Thus, the GAIN dataset cannot be omniscient or cover every aspect, in spite of its rich diversity.  Besides, while causal inference shows its potential to improve generalizability, designing algorithms for generalization is still an open question. We argue that it is worth devoting efforts to this question and leave it as future work.
>
> > **Q7 - Other comments (1):** In Figure 3 (a), there are some prominent spikes. It is interesting to know what are these top frequent steps in COIN Train, COIN Test, and GAIN-C.
>
> **A7:** The prominent spikes are steps with largest frequency rates. The 5 most frequent steps in COIN Train are "wipe the screen", "add different kinds of ingredients", "put on the hair extensions", "mix the raw materials", and "iron the cloths with the iron"; in COIN Test are "put on the hair extensions", "wipe the screen", "dribble in the field", "rise to the sky", and "put down the hair and comb"; and in GAIN-C are "add some water to the tea", "mix the raw materials", "pre-swing", "add raw materials", and "begin to jump up".
>
> > **Q8 - Other comments (2):** Would the causal based method reduce the number of unique steps a video has in predictions (compared to before applying the causal-based method)?
>
> **A8:** We counted the number of unique steps a video has in predictions of MS-TCN++ and C-MS-TCN++ on GAIN-C. The average number of unique steps in predictions of MS-TCN++ is 3.6 and that of C-MS-TCN++ is 3.4. Hence, the numbers of unique steps with the causal based method is slightly smaller than the one with baseline method.

---

> ### Comment · Reviewer_ESEp · 2022-11-20
> **Thank you for the response!**
>
> Most of my questions/concerns were addressed, though I do feel readers might hope to gain more insights from the failure case discussion (currently, not so many insights there, excepts the proposed method is better than existing works).

---

> > ### Author Response · Authors · 2022-11-21
> > **Response to Reviewer ESEp**
> >
> > Thanks for the prompt reply. We are glad that our response and updated presentation have addressed most of your concerns.
> >
> > We greatly appreciate your further helpful feedback. We'll find more failure cases to summarize the principles and provide more underground insights. We are not allowed to modify the paper in the current stage, but we'll update the failure case discussion in the comment, and add it in the final appendix.

---

> > ### Author Response · Authors · 2022-11-23
> > **Further discussion of failure cases**
> >
> > Dear Reviewer ESEp,
> >
> > Thanks again for your reply and additional suggestion for the failure case discussion.
> >
> > Instead of focusing on the videos where both methods fail, we analyze the underlying insights from the cases that the causal-based method has lower performance than the baseline, such as the video "Make Mixed Juice" in the GAIN-C dataset. Take a closer look, we find that there are some strong step dependencies, such as "juice the fruit" and "pour the juice". In this situation, context bias has positive effects on the evaluation. Thus, we got the following insights, despite the better performance on average, our method encourages step independency, which has negative effects on the examples where strong step dependencies occur for both training and testing data. It is reasonable since these steps are in-distribution samples where the context bias has a positive effect, e.g., "juice the fruit" and "pour the juice" are successive steps in both COIN and GAIN-C, so the prediction with the concurrence of them is better. We'll modify the failure case discussion in the appendix.
> >
> > We greatly appreciate your time in reviewing and discussing the rebuttal. Please let us know if further concerns or suggestions remain.

---

> ### Author Response · Authors · 2022-12-01
> **Could you please kindly check whether our follow-up responses properly addressed your concern?**
>
> Dear Reviewer ESEp,
>
> Thanks for your quick reply and additional suggestions. We have provided further discussions for the failure cases. Could you please check whether they properly addressed your concern? We understand you are busy and your feedback would be greatly appreciated. We hope we will have the opportunity for further discussions. Thank you very much!
>
> With best regards,
>
> Authors of submission 1910

---

### Official Review · Reviewer_Ct13 · 2022-10-27

**Confidence:** 3
**Correctness:** 4
**Technical Novelty And Significance:** 3
**Empirical Novelty And Significance:** 3
**Recommendation:** 6

**Clarity, Quality, Novelty And Reproducibility:**

The paper is well-motivated as well as well-written. Each section of the paper is easy to follow. Visualizations and tables are clear and informative.

**Strength And Weaknesses:**

Strength:
1. The GAIN dataset is constructed with great effort. The basic principles to construct datasets are well-designed and in line with the definition of IAU generalizability. Also, the collection process of GAIN is in detailed description. As reported in the experiment section, performance on GAIN-C and GAIN-B undergoes an obvious drop compared to COIN and Breakfast, which validate the effectiveness of benchmarking model’s generalizability on GAIN.
2. Formulation of the causal graph is also clear. The proposed intervention method has solid theoretical proof.

Weakness
1. Action segmentation papers always report metrics of both framewise accuracy and segmental edit distance and the segmental F1 score. It would be nice to see more metrics reported.
2. Models used in the experiment section are all pre-trained ones. I was wondering whether models trained from scratch on datasets (there have been multiple recent works focusing on this) still face a huge drop on GAIN. Also, whether they can be improved by causal-based methods is also unknown.

**Summary Of The Paper:**

The authors introduce the benchmarking evaluation datasets GAIN constructed from the real world, whose videos contain steps from consistent categories with either COIN or Breakfast datasets but are out-of-distribution and belong to non-overlapping task categories. Furthermore, authors make attempts to alleviate performance drop on GAIN by utilizing causal inference and Monte Carlo method to intervene training samples by disassembling and reassembling.

**Summary Of The Review:**

The paper is well motivated and the proposed GAIN dataset has valuable contribution to the future instructional action understanding research. Therefore, my rating to this paper is "6: marginally above the acceptance threshold".

---

> ### Author Response · Authors · 2022-11-16
> **Response to Reviewer Ct13**
>
> Dear Reviewer Ct13, we appreciate your time dedicated to reviewing this paper and providing the valuable comments, which have helped improve our work. We have updated the paper and appendix accordingly. Please see our point-to-point response below.
>
> > **Q1:** Action segmentation papers always report metrics of both framewise accuracy and segmental edit distance and the segmental F1 score. It would be nice to see more metrics reported.
>
> **A1:** Thanks for your valuable suggestion! In light of your suggestion, we further evaluated models with metrics of F1 & edit scores. The performance of baseline methods and their causal versions on Breakfast/GAIN-B are shown in the table below (Please refer to Table 3 and the first two paragraphs in Sec. 4.2 in the revised paper), where A/B denotes that A is the performance on the original Breakfast test set and B is the performance on GAIN-B (the OOD scenario). We found some interesting and reasonable phenomena from these results. First, we found that causal-based methods achieve consistent improvements with both metrics in the OOD scenario, which further demonstrates the effectiveness of the causal method for the OOD problem. Second, we found that the performances of LSTM are poor with these metrics, while other baseline methods work well. By qualitatively checking the predictions, we found that LSTM can only tell actions from the background video content, but fails to classify the action categories correctly. It may be because the LSTM model is overly dependent on temporal relations and weakens the representational capacity.
>
> Table A: Results of 4 baseline methods and corresponding causal version on Breakfast/GAIN-B with edit score and F1 score.
> | Method |  ED-TCN  | C-ED-TCN |  TResNet | C-TResNet | MS-TCN++ | C-MS-TCN++ | LSTM    | C-LSTM  |
> |--------|:--------:|:--------:|:--------:|:---------:|----------|------------|---------|---------|
> | F1@10 (Breakfast/GAIN-B)  | 37.8/3.2 | 42.7/5.3 | 41.6/3.7 |  43.0/5.5 | 57.1/3.1 | 54.7/3.7   | 4.0/0.3 | 4.8/0.5 |
> | Edit (Breakfast/GAIN-B)  | 39.0/4.7 | 44.3/5.9 | 44.1/6.7 |  46.4/8.8 | 54.6/4.9 | 56.4/6.4   | 3.5/1.1 | 6.2/1.8 |
>
> > **Q2:**  Models used in the experiment section are all pre-trained ones. I was wondering whether models trained from scratch on datasets still face a huge drop on GAIN. Also, whether they can be improved by causal-based methods is also unknown.
>
> **A2:** Thanks very much for your kind suggestion! It is interesting to explore the model training from scratch. We have done our best to train the experiments during the rebuttal phase. We trained the models from scratch on COIN for its larger data scale than Breakfast. We're now training MS-TCN++ and C-MS-TCN++ with S3D as backbone and will later evaluate them on COIN/GAIN-C. It still needs weeks for us to finish the training process due to our limited computation resource.
> We will include the results and analysis of this experiment in our appendix.

---

> > ### Author Response · Authors · 2022-12-02
> > **Response to Reviewer Ct13, Part II**
> >
> > Dear Reviewer Ct13,
> >
> > Thanks for your kind suggestion. We just finished the training process due to the limited computation resource. We trained MS-TCN++ and C-MS-TCN++ with S3D as the backbone from scratch on COIN, and we found that both models collapse and only predict the background. It is reasonable because  we used a low sampling rate while training from scratch since the videos in COIN are always long-term. Besides the features extracted from the backbone lack representational capacity without pretraining. We'll modify the experimental settings and hyper-parameters, and try our best to conduct this experiment. Please let us know if further concerns or suggestions remain.
> >
> > With best regards,
> >
> > Authors of submission 1910

---

> ### Author Response · Authors · 2022-12-01
> **Thanks for the valuable suggestions and welcome further feedback**
>
> Dear Reviewer Ct13,
>
> Thanks a lot for your time devoted to reviewing this paper. We have provided responses to your comments and an updated submission. Could you please check whether they properly addressed your concern? We understand you are busy and your feedback would be greatly appreciated. Please kindly let us know in case there are other concerns--we hope we will have the opportunity to respond to them. Thank you very much!
>
> With best regards,
>
> Authors of submission 1910

---

### Author Response · Authors · 2022-11-17
**Response to all reviewers**

Dear Reviewers Ct13, ESEp, and 2UkL:

Thanks for the thoughtful and constructive review. It is encouraging that the reviewers think GAIN is valuable(Reviewer Ct13 and ESEp), novel(Reviewer ESEp), and interesting (Reviewer 2UkL). We here provide a general response to summarize the modifications of the paper.

* To reviewer Ct13: We have added experimental results in Table 3 and provided analysis.
* To reviewer Ct13: We have been training the model from scratch and will include the results in the appendix.
* To reviewer ESEp: We have clarified the assumptions of the causal graph in Sec. 3.1.
* To reviewer ESEp: We have added more comparisons with CrossTask in the introduction.
* To reviewer ESEp: We have added a parameter analysis experiment for the relative size of the augmented data in Sec. C.1.
* To reviewer ESEp: We have clarified the meaning of "category diverse" in Sec.2.2.
* To reviewer ESEp: We have clarified the meaning of "caption clues" in Sec.2.2.
* To reviewer ESEp: We have added failure case analysis with corresponding the visualization in Sec. C.2.
* To reviewer ESEp: We have added a section to discuss limitations in Sec. D.
* To reviewer 2UkL: We have added background knowledge in Sec. 1 for readability.
* To reviewer 2UkL: We have added a description of how to deal with undefined steps in Sec. 2.3.

Thanks again for your time dedicated to carefully reviewing this paper. We hope our response and the change in the paper properly address your concerns.

With best regards, Authors of submission 1910

---

### Decision · Program_Chairs · 2023-01-20

**Decision:**

Accept: poster

**Justification For Why Not Higher Score:**

Reviewers have some concerns regarding the clarity, presentation, and result discussion of this paper. The benchmark is useful but not broadly applicable.

**Justification For Why Not Lower Score:**

The OOD problem is well motivated and formulated. The pipeline for constructing the new OOD evaluation dataset from the existing COIN and Breakfast datasets is also reasonable. In addition, a causality-based method is introduced to improve the generalizability. Overall, the AC agrees with the reviewers that this work would be a valuable benchmark for OOD study for video understanding and thus recommends an accept.

**Metareview: Summary, Strengths And Weaknesses:**

This paper aims to study the out-of-distribution problem for instructional action understanding in videos, and therefore introduces a new benchmark for it. The OOD problem is well motivated and formulated. The pipeline for constructing the new OOD evaluation dataset from the existing COIN and Breakfast datasets is also reasonable. In addition, a causality-based method is introduced to improve the generalizability. The main concerns from the reviewers are about the clarity, presentation, and result discussion (so no critical concerns regarding the task and method formulation). Overall, the AC agrees with the reviewers that this work would be a valuable benchmark for OOD study for video understanding and thus recommends an accept. but we expect that the authors could revise the camera ready version according to the reviews by improving the presentation and adding more result discussion and analysis.

**Note From Pc:**

if the above contains the word "oral" or "spotlight" please see: "oral" presentation means -> notable-top-5% and "spotlight" means -> notable-top-25%. As stated in our emails, we are disassociating presentation type from AC recommendations